# LSP: Low-Power Semi-structured Pruning for Vision Transformers

## Abstract

Vision transformers (ViTs) have emerged as a promising alternative to convolutional neural networks (CNNs) for various image analysis tasks, offering comparable or superior performance. However, one significant drawback of ViTs is their resource-intensive nature, leading to increased memory footprint, computation complexity, and power consumption. To democratize this high-performance technology and make it more environmentally friendly, it is essential to compress ViT models, reducing their resource requirements while maintaining high performance. In this paper, we introduce a new block-structured pruning to address the resource-intensive issue for ViTs, offering a balanced trade-off between accuracy and hardware acceleration. Unlike unstructured pruning or channel-wise structured pruning, block pruning leverages the block-wise structure of linear layers, resulting in more efficient matrix multiplications. To optimize this pruning scheme, our paper proposes a novel hardware-aware learning objective that simultaneously maximizes speedup and minimizes power consumption during inference, tailored to the block sparsity structure. This objective eliminates the need for empirical look-up tables and focuses solely on reducing parametrized layer connections. Moreover, our paper provides a lightweight algorithm to achieve post-training pruning for ViTs, utilizing second-order Taylor approximation and empirical optimization to solve the proposed hardware-aware objective. Extensive experiments on ImageNet are conducted across various ViT architectures, including DeiT-B and DeiT-S, demonstrating competitive performance with other pruning methods and achieving a remarkable balance between accuracy preservation and power savings.

## 1 Introductions

Recently, vision transformers (ViTs) have been an emerging string of research that greatly challenges the prevailing CNNs with on-par or even superior performance on various image analysis and understanding tasks such as classification Dosovitskiy et al. (2020); Cordonnier et al. (2020); Touvron et al. (2021a); Han et al. (2021b); He et al. (2022), object detection Carion et al. (2020); Zhu et al. (2021b); Amini et al. (2021), semantic segmentation Chen et al. (2021a); Liu et al. (2021), etc., but completely without the convolution mechanism seen in the CNNs. Despite the success in the task performances, as pointed out by Yu et al. (2021a), one major drawback of the ViTs architecture is that the ViTs are much less resource-efficient than CNNs in terms of memory footprint, computation complexity and the eventual power consumption. To make the high-performance ViTs more environmental friendly and democratize the technology, it is necessary to compress the ViTs models and cut down the power consumption, so that they could be accessed by low-end computation devices with equal or comparable model performance.

Among different bifurcations of neural network compression, network pruning is an effective method that has shown success on CNNs, which prunes out redundant neurons or rules out computations in the networks. Previously on CNNs, some Han et al. (2015a;b); Zhu & Gupta (2018); Lee et al. (2020); Morcos et al. (2019); Lin et al. (2020); Wang et al. (2022); Xu et al. (2023) attempted *unstructured pruning* to the models which removes individual neurons from the layer weights; while othersLuo et al. (2017); Shen et al. (2022) used *structured pruning* which removes channel-wise neurons. Comparing to unstructured pruning, the latter structured scheme has high data locality hence is more hardware-friendly Buluc & Gilbert (2008) as it is easier to achieve ac-

celerated computation by simply removing entire rows or columns in the weight matrices, it cause severer accuracy degradation due to the coarser pruning granularity making it a much more challenging pruning scheme.

Nevertheless, for transformer architectures consisting of mostly linear layers (matrix multiplication), block structured (semi-structured) pruning is a better trade off between accuracy and hardware acceleration, since the GEMM performs matrix multiplication in a block-by-block manner. Hence multiplication with block sparse matrices can achieve more speedup than unstructured ones under the same pruning ratio while still maintaining high accuracy. A summarized qualitative comparison among pruning schemes is listed in Fig. 1. Prior arts Mao et al. (2021); Lagunas et al. (2021) in NLP domain validated the block structured pruning on language models (BERT , MobileBERT , etc.), achieving more than $2\times$ speedup with negligible performance drop. However, the other parts of their pruning scheme is rather out-dated, *e.g.* vanilla pruning criterion. Similar attempts are still scarce on ViTs for various vision tasks.

| Sparsity scheme | Accuracy | Hardware speedup |
|---|---|---|
| Unstructured | High | Bad |
| Structured | Bad | High |
| Semi-structured (block-sparse) | Good | Good |

Figure 1: Trade-offs of different sparsity schemes in terms of model accuracy and hardware acceleration.

In this work, we propose a novel block-structured pruning approach for ViTs to prune the parameters in a block-based manner to achieve better trade-off between accuracy and efficiency. We formulate the learning objective in a way that simultaneously maintains the accuracy of the pruned model and minimizes the number of the computational operations. A hardware-aware constraint is incorporated into the objective to boost the speedup and lower power consumption during inference stage. Moreover, we present a fast optimization method to solve the objective function by utilizing second-order Taylor approximation. After equivalent reformulation, such we are able to solve the objective very efficiently (quadratic to cubic complexity for empirical data collection against network size and linear time complexity for equation solving). To the best of our knowledge, this is the first paper that introduces the block-structured pruning scheme and present a hardware-aware post-training pruning approach for ViTs. The main contributions are summarized as below:

- We systematically formulate an optimal hardware-aware pruning objective for ViTs models under the block-structured pruning scheme, which directly optimizes both model accuracy and power consumption at the same time. The power consumption is fully estimated without the need of constructing any empirical look-up tables (LUTs), which makes it a light-weight approach and does not require additional overheads for optimization. The proposed pruning scheme solely relies on reducing parametrized layer connections without manipulating skip configurations and token pruning.

- We then provide an efficient solution for the proposed hardware-aware objective function by utilizing second-order taylor approximation and present an empirical optimization method with only linear time complexity. The proposed method firstly generates the curves of the relationships between pruning rate and output error for each layer. Then, it is able to efficiently find the solution under different pruning rates in a fast way and does not need to re-solve the objective function each time when a pruning rate of the model is given.

- Extensive experiments demonstrate the effectiveness of our approach. Results on various deep ViTs architectures, including DeiT-B and DeiT-S, show that our approach noticeably outperforms the state-of-the-arts regarding the trade-off between accuracy and speedup on the ImageNet dataset.

## 2 RELATED WORKS

### 2.1 VISION TRANSFORMERS (VITS)

Following the success of self-attention based transformer architecture in natural language processing Vaswani et al. (2017), transformer based vision models have also been marching in image domain and being strong competitors against traditional CNNs in various scenes like object detection Carion et al. (2020); Zhu et al. (2021b), segmentation Chen et al. (2021a), etc. ViT Dosovitskiy et al. (2021) was the first attempt to introduce MHA (multi-head attention) architecture for image

modality and surpassed the CNNs performance on image classification on large scale datasets. Later, DeiT Touvron et al. (2021b) further boost the performance of raw ViTs with the same architecture but with token-based knowledge distillation to enhance the representation learning. MAE He et al. (2022) introduces a supervision technique to pretrain ViT encoder on masked image reconstruction pretext task and achieves state-of-the-art performance on ImageNet classification task. Swin Transformer Liu et al. (2021) utilized shifted window to introduce inter-window information exchange and enhance local attention. Transformer-iN-Transformer (TNT) Han et al. (2021a) aggregated both patch- and pixel-level representations by a nested self-attention within each transformer block.

## 2.2 PRUNING ON CNNS

CNNs pruning has been widely studied for decades. Large amount of pruning methods can be categorized in many different way. Depending on the relationship between pruning and training procedure, they can be divided into post-train-pruning, pruning-at-initialization and pruning-during-training, where this work falls into post-train-pruning scheme as we determine the pruning mask on a converged pretrained model. Depending on the the level of sparsity, it can be grouped into unstructured pruning, semi-structured pruning, structured (channel/filter-wise) pruning, etc. We introduce the related works base on the later taxonomy.

**Unstructured Pruning** removes individual connections (neurons) from convolution kernels, which is the earliest established pruning scheme by the pioneer works Han et al. (2015a;b), where they adopt a magnitude-base criterion with iterative fine-tuning procedure for LeNet and AlexNet. Molchanov et al. (2016) adopted taylor-based criterion as a importance score for connection. Frankle & Carbin (2019) proposed the lottery hypothesis deriving a weight-rewinding technique in iterative-pruning. Morcos et al. (2019); Zhu & Gupta (2018) adopts magnitude-based importance scores to threshold low-scored connections globally. Gale et al. (2019); Evci et al. (2020) leverage architectural heuristics to determine layerwise pruning rate. Lee et al. (2020) improved the magnitude-based scores like in Morcos et al. (2019) by considering inter-layer score ranking. Several efforts also prune CNNs data-dependently, considering the influence of pruning on the model output. Molchanov et al. (2016); Lee et al. (2019) derived a first-order taylor-based pruning criterion. Isik et al. (2022) assumed laplacian distribution of CNN weights to approximate output distortion to determine layer-wise pruning ratio. Wang et al. (2022); Xu et al. (2023) leverage rate-distortion theory to derive layer-wise pruning ratios that achieves optimal rate-distortion performance. Unstructural pruning achieves minimal sparse model accuacy thanks to the most fine-grained sparsity pattern, but such irregular sparsity pattern unfortunately makes it hard to achieve real-world acceleration without dedicated hardware optimization due to the poor data locality and low parallelism.

**Structured Pruning** or channel/filter-wise pruning scheme prunes the entire kernel in a Conv layer or a channel in fully connected layer at once. Luo et al. (2017) used feature map importance as a proxy to determine removable channels. He et al. (2017) took a regularization based strucrtural pruning method. Yu et al. (2018) obtains channel-wise importance scores by propagating the score on the final response layer. Lin et al. (2020) utilized rank informtation of feature maps to determine the prunable channels. Wang et al. (2022) leveraged rate-distortion theory to prune the channels that lead to least model accuracy drop. Shen et al. (2022) took first-order importance on channels and allocates sparsities by solving a knapsack problem on all channel importances in the whole network. Structured pruning adopts coarser sparsity pattern than unstructured pruning, which trades-off the model accuracy with easily achievable acceleration.

**Semi-Structured Pruning** is a less-explored approach that leverages sparsity pattern in between unstructured and structured pruning, where patterns such as block-sparsity in matmul can greatly benefit the realworld speedups by exploiting the nature of GPU calculation Mao et al. (2021); Lagunas et al. (2021). With the sparsity pattern less agressive than sturctured pruning, the impact of removing neurons on the model accuracy is less than structured pruning. Nevertheless, semi-structured pruning is under-explored on the emerging ViTs, which are constructured with transformer encoder architecture with mostly fully connected layers.

## 2.3 SPARSITY IN VITS

Witnessing the success of CNNs pruning, ViTs pruning is also receiving emerging interests. Compared to CNNs pruning, less efforts are devoted to pure weight pruning but more on pruning of

tokens, MHA, etc. S$^2$ViTE Chen et al. (2021b) first proposed to prune out tokens as well as self-attention heads under structured pruning scheme with sparse training for ViTs. UVC Yu et al. (2021b) derived a hybrid optimization target that unifies structural pruning for ViT weights, tokens and skip configuration to achieve sparse training for ViTs. SPViT Kong et al. (2022) only performed token pruning on attention heads but adopted latency constraint to maximize speedup on edge devices. Yang et al. (2023) adopts Nvidia's Ampere 2:4 sparsity structure to achieve high speedup but required structural constraints to ensure a matching dimensions of qkv, feedforward and projection layers (head alignment) to search for subnetwork from larger ViT variants to match the latency of smaller ones. Unlike prior works Yu et al. (2021a); Yang et al. (2023), our method focuses on pure weight pruning scheme and does not require heavy searching for the coordination of different compression schemes. Some efforts Kitaev et al. (2019); Wu et al. (2019); Wang et al. (2021); Zaheer et al. (2020) sparsify the heavy self-attention by introducing sparse and local attention patterns for language models. Child et al. (2019) attempts on ViTs, but these sparse attention schemes still require training from scratch.

## 3 METHODOLOGIES

### 3.1 PRELIMINARIES

**Block-structured Pruning within layer.** We targeted at block-structured pruning for all linear layer weights, which include any parametrized linear layers in the ViTs, such as qkv layers, feedforword and projection layers. Neurons in these weight matrices are grouped in 2-dimensional fixed-sized blocks as a unit for pruning. To decide which blocks need to be pruned, given a block structure $(B_h, B_w)$, for each matrix $\boldsymbol{W} \in \mathbb{R}^{H \times W}$, we rank the blocks by the average of 1st order taylor expansion score of the neuron within each block. Mathematically, we first obtain the neuron score by the taylor expansion $\boldsymbol{S} = |\boldsymbol{W} \cdot \nabla_{\boldsymbol{W}} f|$ similar to Molchanov et al. (2019), then perform a 2D average pooling to obtain a score for each block $\boldsymbol{S}' \in \mathbb{R}_*^{H/B_h \times W/B_w}$ ($\mathbb{R}_*$ is non-negative real value set). Then given a pruning ratio for each layer, we can rank the blocks by their scores and eliminate the bottom ranked ones. The right most part of Fig. 2 visualizes the block-structure patterns realistically generated from ViTs. The above pruning scheme can be formulated as $\tilde{\boldsymbol{W}}_{i,j} = \boldsymbol{W}_{i,j} \odot \boldsymbol{M}_\alpha(\boldsymbol{S}')_{\lceil \frac{i}{B_h} \rceil, \lceil \frac{j}{B_w} \rceil}$, where $\boldsymbol{M}_\alpha(\boldsymbol{S}')$ is the binary mask generated from the previous block-wise score matrix under the pruning ratio $\alpha$.

**Pruning scheme of ViTs.** Unlike prior arts, the scope of this work is only eliminating model parameters to reduce computation, without considering other aspects of ViTs like token number and token size and transformer block skipping Chen et al. (2021b); Yu et al. (2021b); Kong et al. (2022).

We further adopt a basic assumption for the weight perturbation $\Delta \boldsymbol{W} = \tilde{\boldsymbol{W}} - \boldsymbol{W}$ caused by a typical pruning operation to the weight:

**Assumption 1** *I.i.d. weight perturbation across layers* Zhou et al. (2018): *This means the joint distribution is zero-meaned:* $\forall 0 < i \neq j < L, E(\Delta \boldsymbol{W}^{(i)} \Delta \boldsymbol{W}^{(j)}) = E(\Delta \boldsymbol{W}^{(i)}) E(\Delta \boldsymbol{W}^{(j)}) = 0$, *and also zero co-variance:* $E(\|\Delta \boldsymbol{W}^{(i)} \Delta \boldsymbol{W}^{(j)}\|^2) = 0$.

### 3.2 HARDWARE-AWARE PRUNING OBJECTIVE

Since layers may contribute differently to the model performance Frankle et al. (2020), various criteria have been proposed to allocate layerwise sparsity given a total budget . However, most existing pruning objectives can be summarized as minimizing the model output accuracy under computation constraint, without explicitly taking into account the actual power consumption and speedup. In contrast, our compression pruning objective directly optimizes the power consumption to achieve certain computation reduction target (FLOPs). Specifically, given a neural network $f$ of $l$ layers and its parameter set $\boldsymbol{W}^{(1:l)} = \left(\boldsymbol{W}^{(1)}, ..., \boldsymbol{W}^{(l)}\right)$, where $\boldsymbol{W}^{(i)}$ is the weights in layer $i$, pruning parameters in the $f$ will give a new parameter set $\tilde{\boldsymbol{W}}^{(1:l)}$. We view the impact of pruning as the distance between the network outputs $f(x; \boldsymbol{W}^{(1:l)})$ and $f(x; \tilde{\boldsymbol{W}}^{(1:l)})$.

Hence our learning objective is as follows:

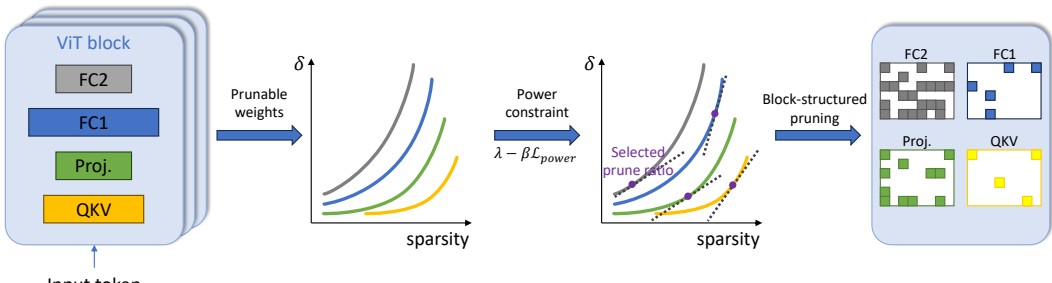

Figure 2: Illustration of the proposed Low Power Semi-structured pruning method. Widths of different layers within ViT block visualizes the computation complexities (FLOPs) of single layer. We first extract all layers with prunable weights in the pretrained ViTs, then we obtain the empirical curves $\delta$-vs-sparsity as described in Eq. 11. We further calculate the layer specific target slope $\lambda_i$ according to its contribution to the power consumption and select the layer-wise pruning ratios when the target slopes are tangential to the curves. Finally we prune the layer weights given their pruning ratios in block-structured sparsity, and finally finetune the pruned ViTs. The rightmost of the diagram is an example of the block-sparsity patterns when block sizes for both dimensions are the same, but they don't have to be the same as in the experiment section.

$$\min \quad \|f(x; \boldsymbol{W}^{(1:l)}) - f(x; \tilde{\boldsymbol{W}}^{(1:l)})\|^2 + \beta\mathcal{L}_{power}(f(\tilde{\boldsymbol{W}}^{(1:l)})) \quad s.t. \frac{\text{FLOPs}(f(\tilde{\boldsymbol{W}}^{(1:l)})}{\text{FLOPs}(f(\boldsymbol{W}^{(1:l)})} \leq R,$$

(1)

which jointly minimize the output distortion caused by pruning (first term) as well as the estimated power consumption $\mathcal{L}_{power}(f(\tilde{\boldsymbol{W}}^{(1:l)}))$, under a certain FLOPs reduction target $R$.

### 3.3 SECOND-ORDER APPROXIMATION OF OUTPUT DISTORTION

To solve the pruning objective, we break down the first term related to the output distortion. We first expand the output distortion $f(x; \boldsymbol{W}^{(1:l)}) - f(x; \tilde{\boldsymbol{W}}^{(1:l)})$ using second-order taylor expansion: (omit the superscript $(1:l)$ for visual clarity from now)

$$f(x; \boldsymbol{W}) - f(x; \tilde{\boldsymbol{W}}) = \sum_{i=1}^{l} \nabla_{\boldsymbol{W}^{(i)\top}} f \Delta \boldsymbol{W}^{(i)} + \frac{1}{2}\Delta \boldsymbol{W}^{(i)\top} \boldsymbol{H}_i \Delta \boldsymbol{W}^{(i)},$$

(2)

where $\boldsymbol{H}_i$ is the hessian matrix of the $i$-th layer weight.

Then consider the expectation of the squared L2 norm in the objective Eq. 1, which can be rewritten as the vector inner-product form:

$$E(\|f(x; \boldsymbol{W}) - f(x; \tilde{\boldsymbol{W}})\|^2) = E\left[(f(x; \boldsymbol{W}) - f(x; \tilde{\boldsymbol{W}})^\top (\|f(x; \boldsymbol{W}) - f(x; \tilde{\boldsymbol{W}})\right]$$

$$= \sum_{i,j=1}^{l} E\left[\left(\nabla_{\boldsymbol{W}^{(i)}}^\top f \Delta \boldsymbol{W}^{(i)} + \frac{1}{2}\Delta \boldsymbol{W}^{(i)\top} \boldsymbol{H}_i \Delta \boldsymbol{W}^{(i)}\right)^\top \left(\nabla_{\boldsymbol{W}^{(j)}}^\top f \Delta \boldsymbol{W}^{(j)} + \frac{1}{2}\Delta \boldsymbol{W}^{(j)\top} \boldsymbol{H}_j \Delta \boldsymbol{W}^{(j)}\right)\right].$$

(3)

When we further expand the inner-product term, the cross-term for each pair of different layer $1 \leq i \neq j \leq l$ is:

$$E\left[\Delta \boldsymbol{W}^{(i)\top} \nabla_{\boldsymbol{W}^{(i)}} f \nabla_{\boldsymbol{W}^{(j)}}^\top f \Delta \boldsymbol{W}^{(j)}\right] + E\left[\frac{1}{2}\Delta \boldsymbol{W}^{(i)} \Delta \boldsymbol{W}^{(i)\top} \boldsymbol{H}_i^\top \nabla_{\boldsymbol{W}^{(j)}}^\top f \Delta \boldsymbol{W}^{(j)}\right] +$$

$$E\left[\frac{1}{2}\Delta \boldsymbol{W}^{(i)\top} \nabla_{\boldsymbol{W}^{(i)}} f \Delta \boldsymbol{W}^{(j)\top} \boldsymbol{H}_j \Delta \boldsymbol{W}^{(j)}\right] + E\left[\frac{1}{4}\Delta \boldsymbol{W}^{(i)} \Delta \boldsymbol{W}^{(i)\top} \boldsymbol{H}_i^\top \Delta \boldsymbol{W}^{(j)\top} \boldsymbol{H}_j \Delta \boldsymbol{W}^{(j)}\right].$$

(4)

When we dicuss the influence of the random variable $\Delta \boldsymbol{W}$, the first-order and second-order derivatives $\nabla_{\boldsymbol{W}} f$ and $\boldsymbol{H}$ can be regarded as constants and therefore can be moved out of expectation. Also vector transpose is agnostic inside expectation. So Eq. 4 becomes

$$\nabla_{\boldsymbol{W}^{(i)}} f \nabla_{\boldsymbol{W}^{(j)}}^\top f E(\Delta \boldsymbol{W}^{(i)\top} \Delta \boldsymbol{W}^{(j)}) + \frac{1}{2}\boldsymbol{H}_i^\top \nabla_{\boldsymbol{W}^{(j)}}^\top f E(\Delta \boldsymbol{W}^{(i)} \Delta \boldsymbol{W}^{(i)\top} \Delta \boldsymbol{W}^{(j)}) +$$

$$\frac{1}{2}\nabla_{\boldsymbol{W}^{(i)}} f \boldsymbol{H}_j E(\Delta \boldsymbol{W}^{(i)\top} \Delta \boldsymbol{W}^{(j)\top} \Delta \boldsymbol{W}^{(j)}) + \frac{1}{4}\boldsymbol{H}_i^\top \boldsymbol{H}_j E(\|\Delta \boldsymbol{W}^{(i)\top} \Delta \boldsymbol{W}^{(j)}\|^2).$$

(5)

Using Assumption 1, we can find that the above 4 cross-terms also equal to zero [1]. Therefore the expectation Eq. 3 results in only intra-layer terms:

$$E(\|f(x; \boldsymbol{W}) - f(x; \tilde{\boldsymbol{W}})\|^2) = \sum_{i=1}^{l} E\left(\left\|\nabla_{\boldsymbol{W}^{(i)}}^{\top} f \Delta \boldsymbol{W}^{(i)} + \frac{1}{2} \Delta \boldsymbol{W}^{(i)\top} \boldsymbol{H}_i \Delta \boldsymbol{W}^{(i)}\right\|^2\right). \quad (6)$$

### 3.4 POWER CONSUMPTION UNDER BLOCK-STRUCTURED PRUNING

As the majority of the power consumption of network inference is attributed to the matrix multiplication operation, the network power consumption can be estimated by summing individual power cost of block-sparse matrix multiplication of each linear layers. Consider a matrix $\boldsymbol{A} \in \mathbb{R}^{M \times N}$, typically input tensor, to be multiplied with the block-sparse weight matrix $\boldsymbol{B} \in \mathbb{R}^{N \times K}$ with block-structure of $(B_n, B_k)$ and $\alpha$-percentage of blocks pruned out. When using a block-sparse GEMM configured with the kernel grid size of $B_m$ on $M$-dimension, the power consumption of the block-sparse matmul can be estimated as

$$P = p_m \frac{M}{B_m} \left\lceil (1 - \alpha) \frac{N}{B_n} \frac{K}{B_k} \right\rceil, \quad (7)$$

where $p_m$ is the power cost of individual within-block matmul. Therefore, the second term in Eq. 1 can be obtained by adding up the power consumption of the network of all layers:

$$\beta \mathcal{L}_{power} = \beta p_m \sum_{i=1}^{l} \frac{M_i}{B_m} \left\lceil (1 - \alpha_i) \frac{N_i}{B_n} \frac{K_i}{B_k} \right\rceil, \quad (8)$$

where $p_m$ and $B_m$ can be absorbed into the weight coefficient $\beta$ because they only depends on hardware parameters and GEMM configuration which is unified across layers.

**Final Objective.** Combining Eq. 6 and Eq. 8, the final objective can be reformulated as:

$$\min \sum_{i=1}^{l} E\left(\left\|\nabla_{\boldsymbol{W}^{(i)}}^{\top} f \Delta \boldsymbol{W}^{(i)} + \frac{1}{2} \Delta \boldsymbol{W}^{(i)\top} \boldsymbol{H}_i \Delta \boldsymbol{W}^{(i)}\right\|^2\right) + \beta \sum_{i=1}^{l} M_i \left\lceil (1 - \alpha_i) \frac{N_i}{B_n} \frac{K_i}{B_k} \right\rceil$$
$$s.t. \frac{\text{FLOPs}(f(\tilde{\boldsymbol{W}}^{(1:l)})}{\text{FLOPs}(f(\boldsymbol{W}^{(1:l)})} \leq R. \quad (9)$$

### 3.5 FINDING SOLUTION TO PRUNING OBJECTIVE

At this point, we can further solve the optimization problem Eq. 9 on the layer-wise pruning ratio set $\{\alpha_i \mid 1 \leq i \leq l\}$ by applying lagrangian formulation Wang et al. (2022); Xu et al. (2023)

$$\frac{\partial}{\partial \alpha_i} \left(\nabla_{\boldsymbol{W}^{(i)}}^{\top} f \Delta \boldsymbol{W}^{(i)} + \frac{1}{2} \Delta \boldsymbol{W}^{(i)\top} \boldsymbol{H}_i \Delta \boldsymbol{W}^{(i)} + \beta M_i \left\lceil (1 - \alpha_i) \frac{N_i}{B_n} \frac{K_i}{B_k} \right\rceil\right) = \lambda. \quad (10)$$

In practice we can get rid of the ceiling function in Eq. 10 and therfore:

$$\frac{\partial}{\partial \alpha_i} \left(\nabla_{\boldsymbol{W}^{(i)}}^{\top} f \Delta \boldsymbol{W}^{(i)} + \frac{1}{2} \Delta \boldsymbol{W}^{(i)\top} \boldsymbol{H}_i \Delta \boldsymbol{W}^{(i)}\right) = \lambda_i = \lambda + \beta \frac{M_i N_i K_i}{B_n B_k}, \quad (11)$$

which will give a continuous $\alpha_i \in [0, 1]$ compared to the original solution with the ceiling, but in practice since the number of blocks within a weight tensor is limited the pruning ratio $\alpha_i$ is to be rounded to a discrete value anyway. Solving Eq. 11 will need to collect empirical curves for all layers (pruning ratio $\alpha_i$ against the taylor second-order term $\delta_i = \nabla_{\boldsymbol{W}^{(i)}}^{\top} f \Delta \boldsymbol{W}^{(i)} + \frac{1}{2} \Delta \boldsymbol{W}^{(i)\top} \boldsymbol{H}_i \Delta \boldsymbol{W}^{(i)}$). By setting a specific $\lambda$, we can solve Eq. 11 individually for each layer by searching for a $\alpha_i$ that let the equality holds. The final solution of pruning ratios can be obtained by traversing $\lambda$ that returns a pruned network closest to the constraint $R$.

One *key insight* that one can derive from the optimization solution Eq. 11 is that by controlling the weight $\beta$, the power consumption are explicitly incorporated in the optimization process in the form of altering the target slope for the partial derivative of the curve $\frac{\partial \delta_i(\alpha_k)}{\partial \alpha_k}$, which represents how intensely pruning one layer affects the final model accuracy (output distortion). In this way, we achieve direct tradeoff between model accuracy and power consumption.

---

[1] We empirically find $E(\boldsymbol{W}^{(i)\top} \boldsymbol{W}^{(i)} \boldsymbol{W}^{(j)}) = 0$ holds on top of $E(\boldsymbol{W}^{(i)} \boldsymbol{W}^{(j)}) = 0$.

### 3.6 EMPIRICAL COMPLEXITY

**Hessian approximation.** For empirical networks, we approximate the hessian matrix $\boldsymbol{H}_i$ using *empirical fischer* Kurtic et al. (2022):

$$\boldsymbol{H}_i = \boldsymbol{H}_{\mathcal{L}}(\boldsymbol{W}^{(i)}) \approx \hat{\boldsymbol{F}}(\boldsymbol{W}^{(i)}) = \kappa \boldsymbol{I}_d + \frac{1}{N}\sum_{n=1}^{N} \nabla_{\boldsymbol{W}^{(i)}} f_n \nabla_{\boldsymbol{W}^{(i)}}^{\top} f_n. \tag{12}$$

In order to obtain empirical curves $\frac{\partial \delta_i(\alpha_k)}{\partial \alpha_k}$ on a calibration set, one is possible to traverse different pruning ratio (*e.g.* in practice $\alpha_k = \frac{k+1}{K}, 0 < k < K$) and caluculate the corresponding $\delta_i(\alpha_k)$ for all $0 < k < K$. However in such case, even with the approximated hessian, the curve generation for each layer is still very expensive at the complexity of $O(NKD_i^4)$, where $K$ is the number of possible pruning ratio selections and $D_i = N_i K_i$ is the dimension of weight in $i$-th layer. This poses challenge to make the proposed method efficient enough to enjoy the benefits of sparse network. We notice that the derivative $\nabla_{\boldsymbol{W}_i}$ is constant to the change of pruning ratio which let us to reuse the hessian matrix for all pruning ratio, which drops the complexity to $O((N+K)D_i^2 + KD_i^4)$. However, the existence of the biquadratic complexity makes it still too expensive. We further notice that when pruning ratio move up slightly, only a partition of the weight vector is pruned out from $\tilde{\boldsymbol{W}}_i$. Therefore we can select a subvector $\mathrm{d}\Delta\boldsymbol{W}_i(\alpha_k) = \Delta\boldsymbol{W}_i(\alpha_k) - \Delta\boldsymbol{W}_i(\alpha_{k-1})$ each time when pruning ratio increases from $\alpha_{k-1}$ to $\alpha_k$ and update the $\delta_i(\alpha_k)$ from $\delta_i(\alpha_{k-1})$ by the following rule:

$$\delta_i(\alpha_k) - \delta_i(\alpha_{k-1}) = \nabla_{\boldsymbol{W}^{(i)}}^{\top\prime} f \mathrm{d}\Delta\boldsymbol{W}_i(\alpha_k) + \left(\frac{1}{2}\mathrm{d}\Delta\boldsymbol{W}_i(\alpha_k) + \Delta\boldsymbol{W}_i(\alpha_{k-1})\right)^{\top} \boldsymbol{H}_i' \mathrm{d}\Delta\boldsymbol{W}_i(\alpha_k). \tag{13}$$

Denote the dimension of the subvector $\mathrm{d}\Delta\boldsymbol{W}_i(\alpha_k)$ as $d_i(k) \ll D_i$ equals the number of values changes from $\Delta\boldsymbol{W}_i(\alpha_{k-1})$ to $\Delta\boldsymbol{W}_i(\alpha_k)$, the multiplication calculation in Eq. 13 can be operated at lower dimensions, where $\nabla_{\boldsymbol{W}^{(i)}}^{\top\prime} f \in \mathbb{R}^{d_i(k)}, \boldsymbol{H}_i' \in \mathbb{R}^{D_i \times d_i(k)}$ are subvector and submatrix indexed from the original ones. At $k=1, \alpha_k = 0$ *i.e.* there is no pruning at all which guarantees $\delta_i(\alpha_1) = 0$. Therefore, the complexity becomes one time calculation of the hessian $O(ND_i^2)$ at $k=1$, in addition to $K-1$ times of updating $O(D_i^2 \sum_{k=1}^{K-1} d_i(k)^2)$, resulting in totally $O((N+\sum_{k=1}^{K-1} d_i(k)^2)D_i^2)$ ($d_i(k) \ll D_i$ when $K$ is big enough).

To this end, we presented a hardware-aware pruning criterion that explicitly accounts for the power consumption of the block-structured sparse model inference. The block-structured pruning scheme enables the obtained sparse network to achieve real-world acceleration on hardware while optimally preserving the network accuracy. The algorithm is extremely efficient to obtain a sparse ViT.

## 4 EXPERIMENTS

### 4.1 EXPERIMENT SETTINGS

We conduct experiments mainly on Deit-Small and Deit-Base Touvron et al. (2021b) on ImageNet dataset Krizhevsky et al. (2012). We adopt the same training settings as in UVC Yu et al. (2021a) for the finetuning of ViTs, *e.g.* 300 epochs and the additional distillation token for knowledge distillation. We select 2000 training samples to form the calibration set to calculate the first and second-order derivatives.

**Automatic hyperparameters setting**. As introduced in Sec. 3.5, there are two hyperparameters $\lambda$ and $\beta$ involved in the solution, but both can be adaptively configured without the need to set manually. For the identification of $\beta$, we follow the below strategy:

$$\beta = \frac{\sum_{l=1}^{L} \max_i \frac{\partial \delta_i}{\partial \alpha_i}}{\sum_{l=1}^{L} \max_i \frac{\partial \mathcal{L}_{power}}{\partial \alpha_i}}, \tag{14}$$

so that the scales of the output distortion term and power term is balanced. After the $\beta$ is fixed, we assume the FLOPs of the pruned model is a monotonic function of $\lambda \in [0, \infty)$ and therefore can perform the efficient binary search towards the target FLOPs to obtain the choice of $\lambda$.

**Post-processing of empirical $\delta$ curves.** Due to the high granularity in the block-sparsity structure, the layer-wise $\delta$ curves are expected to see some quantization effect, where the $\delta$ values remains the same corresponding to small change in pruning ratio $\alpha$. This effect is even more severe under larger

block shapes, *e.g.* $64 \times 64$. To better aid the pruning ratio searching procedure, we adopt several post-processing tricks to the empirical curves: (1) **Curve Smoothing**: we perform Exponential Moving Average (EMA) smoothing on the curves. (2) **Curve Derivative Numerical Approximation**: We further approximate the derivatives of $\delta$ curves using 5-point centered difference Sauer (2011) to compared with the target slope (RHS of Eq. 11).

**Baseline methods.** For the following experiments, we followed the UVC Yu et al. (2021a) comparison settings and compare ourselves to the previous ViTs compression methods that at least involves model weights pruning, as well as hybrid methods, including SCOP Tang et al. (2020), VTP Zhu et al. (2021a), S$^2$ViTE Chen et al. (2021b) and UVC Yu et al. (2021a) itself.

## 4.2 MAIN RESULTS

Table 1: Comparisons with state-of-the-art ViTs pruning methods.

| Model | Method | Top-1 Acc (%) | FLOPs(G) | FLOPs remained (%) |
|---|---|---|---|---|
| | Dense | 79.8 | 4.6 | 100 |
| | SCOP | 77.5 (-2.3) | 2.6 | 56.4 |
| Deit-Small | S$^2$ViTE | 79.22 (-0.58) | 3.14 | 68.36 |
| | UVC | 78.82 (-0.98) | 2.32 | 50.41 |
| | LSP (Ours) | **80.69(+0.89)** | 2.3 | 50 |
| | Dense | 81.8 | 17.6 | 100 |
| | S$^2$ViTE | 82.22 (+0.42) | 11.87 | 66.87 |
| Deit-Base | VTP | 80.7 (-1.1) | 10 | 56.8 |
| | UVC | 80.57 (-1.23) | 8 | 45.5 |
| | LSP (Ours) | **80.81($-$0.99)** | 8.8 | 50 |
| | LSP (Ours) | 80.55 (-1.25) | 7.92 | 45 |

As presented in Tab. 1, we first notice that our result on Deit-Small achieves loss-less, and even higher than dense model performance by $0.89$, with roughly the same FLOPs, surpassing existing baselines by a **large margin**. On larger architectures like DeiT-Base, where our method displays less prominent improvement but still on-par performance on the Top-1 accuracy of $80.81$ with $50\%$ FLOPs remaining and $80.55$ with around $45\%$ FLOPs. This is an intuitive observation since coarser pruning patterns like structural pruning would hurt the performance of smaller models more than larger model with a lot more redundant weights, and that is also where smaller structures such as the proposed block-sparsity pattern will retain more performance while still ensure speedup compared to unstructured pruning. Benefit from the pruning scheme tailored to ViTs, we managed to cut down the computation of DeiT-Small by $50\%$ while still have $3\%$ accuracy gain from CNN pruning scheme SCOP Tang et al. (2020) even when they removed slightly less computations ($56.4\%$). We also notice that pure weight pruning of ViTs still have the potentials to achieve superior performance to hybrid methods Chen et al. (2021b); Yu et al. (2021a), thanks to our layer-wise sparsity allocation algorithm that is formulated to directly minimizes the output error on the pruned model against the dense model. On DeiT-Small, we beat all existing hybrid methods that leverage patch-slimming or token selections. We remain competitive on larger model DeiT-Base, while we notice S$^2$ViTE cannot achieve comparable FLOPs reduction to us.

## 4.3 DISCUSSIONS

Beyond the main results, we also attempt to discover how each creative parts in our proposed pruning scheme contribute to the final results, *e.g.* the essential objective constraint regulating the power consumption and the block-sparsity structure, and answering the important questions such as why does the power constraint benefits the performance. We present the detailed ablation studies in the Tab. 2 and Tab. 3.

**Power constraint.** To look deep into how our proposed power efficient pruning scheme accomplishes the above performance gain, we compared the behaviors of our pruning objective with and without the second-term power consumption in Eq. 9. As shown in Tab. 2, we notice that when the FLOPs reduction rates for different settings are both approaching the target FLOPs $50\%$ with only little fluctuations, our final pruning scheme (with power constraint) constantly gives significant higher finetuning results under different block shapes. Specifically, on DeiT-Base-BK32BN64, the

Table 2: Ablation studies on the Power consumption constraint on the pruning result. We compare between the results with the power constraint (main results) and without (by setting $\beta = 0$).

| Method | Acc (%) | Params remained (%) | FLOPs remained (%) |
|---|---|---|---|
| Deit-Base-BK32BN32 | | | |
| w/ Power constraint | 80.81 | 73.3 | 52.5 |
| w/o Power constraint | 77.75 | 26.9 | 55.6 |
| Deit-Base-BK32BN64 | | | |
| w/ Power constraint | 80.71 | 72.8 | 50 |
| w/o Power constraint | 61.42 | 49 | 49.7 |

Table 3: Effects of different Block shape configurations on the pruning result.

| Model | Block shape (BK × BN) | Sparsity (%) | Top-1 Acc (%) | FLOPs remained (%) |
|---|---|---|---|---|
| Deit-Small | $16 \times 16$ | 92.2 | 80.69 | 50 |
| | $32 \times 16$ | 91 | 79.09 | 50 |
| | $16 \times 32$ | 71.37 | 78.2 | 50 |
| | $32 \times 32$ | 49 | 73.32 | 50 |
| Deit-Base | $32 \times 32$ | 72.84 | 80.81 | 52.5 |
| | $64 \times 32$ | 33.93 | 80.05 | 50 |
| | $32 \times 64$ | 16.99 | 80.71 | 50 |
| | $64 \times 64$ | 73.34 | 79.52 | 50.2 |

performance drops by $19.29\%$ when we only remove the power term (setting $\beta = 0$). This is an inspiring phenomenon since the power constraint are not designed to facilitate model accuracy at the first place. By inspecting the model sparsity (number of parameters remained), we learn that the proposed power constraint looks for layers with larger matmul dimensions to allocate more pruning quota to achieve the most impact on the computation reduction (FLOPs), and normally larger layers have more parameter redundancy. Therefore, this pruning ratio allocation actually cooperates with the main objective to minimize output distortion. For both block sizes, model sparsities are far less when the power constraint is removed, *i.e.* at $26.9\%$ and $49\%$ respectively.

**Block structure configurations.** To evaluate how our optimization scheme adapts to different block size configurations, which is crucial to generalize on different hardware platforms with different levels of parallelism, we conducted an ablation studies varying different block shapes combinations as listed in Tab. 3. Firstly, although our algorithm provides around the same FLOPs remaining percentage for different block sizes, it is observed on both test transformer variants that smaller block sizes preserve more model accuracy after finetuning. On Deit-Base, smallest block (BK32BN32) generates the highest $80.81$ accuracy while the largest block (BK64BN64) performs slightly worse at $79.52\%$. On smaller network DeiT-Small, the performance discrepancy is more pronounced, where the largest and smallest block sizes produces the Top-1 accuracy difference at around $7\%$. Secondly, we notice that smaller networks are more sensitive to the change of block shapes. Despite BK32BN32 configuration behaves remarkably on DeiT-Base, the finetuning process for DeiT-Small with BK32BN32 only give $73.32\%$ accuracy with much struggle. By only changing one dimension of the block structure to half size, *e.g.* BK32BN32 to BK32BN16, the performance climbs back by a large margin, returning to acceptable range. Different block sizes results in drastic change to the resulting number of parameters left in the networks, *e.g.* from $92.2\%$ of sparsity on Deit-Small-BK16BN16 to $49\%$ on Deit-Small-BK32BN32.

## 5 CONCLUSIONS

In this work, we presented a novel ViTs weight pruning algorithm designed to reduce energy consumption during inference. Leveraging the linear layer-centric structure of the ViT architecture, we introduced a semi-structured pruning scheme to balance finetuning stability and hardware efficiency. Our algorithm is very efficient despite employing a hessian-based pruning criterion. Experimental results on various ViTs on ImageNet showcase the method's ability to identify optimal pruning solutions, maximizing accuracy for block-sparse models. Additionally, we illustrated the dual benefits of our proposed power-aware pruning objective, enhancing both software accuracy and hardware acceleration.

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

## A    SWIN TRANSFORMERS

To evaluate the effectiveness of LSP on non-global attention transformers such as Swin Transformers Liu et al. (2021), we further conducted experiments on two variants of Swin Transformers. As shown in Tab. 4, LSP remains competitive on Swin Transformer compared to other ViT pruning methods, achieving only 1.96% loss on Swin-Tiny with FLOPs 71.7%.

Table 4: Pruning results on Swin Transformers on ImageNet-1k.

| Model | Method | FLOPs Remained (%) | Blocksize | Top-1 Accuracy |
|---|---|---|---|---|
| Swin-Base | Dense | 100 | - | 83.5 |
| | LSP | 50 | $32 \times 32$ | 79.6 |
| Swin-Tiny | Dense | 100 | - | 81.2 |
| | X-Pruner Yu & Xiang (2023) | 71.1 | / | 78.55 |
| | LSP | 71.1 | $16 \times 16$ | 79.24 |

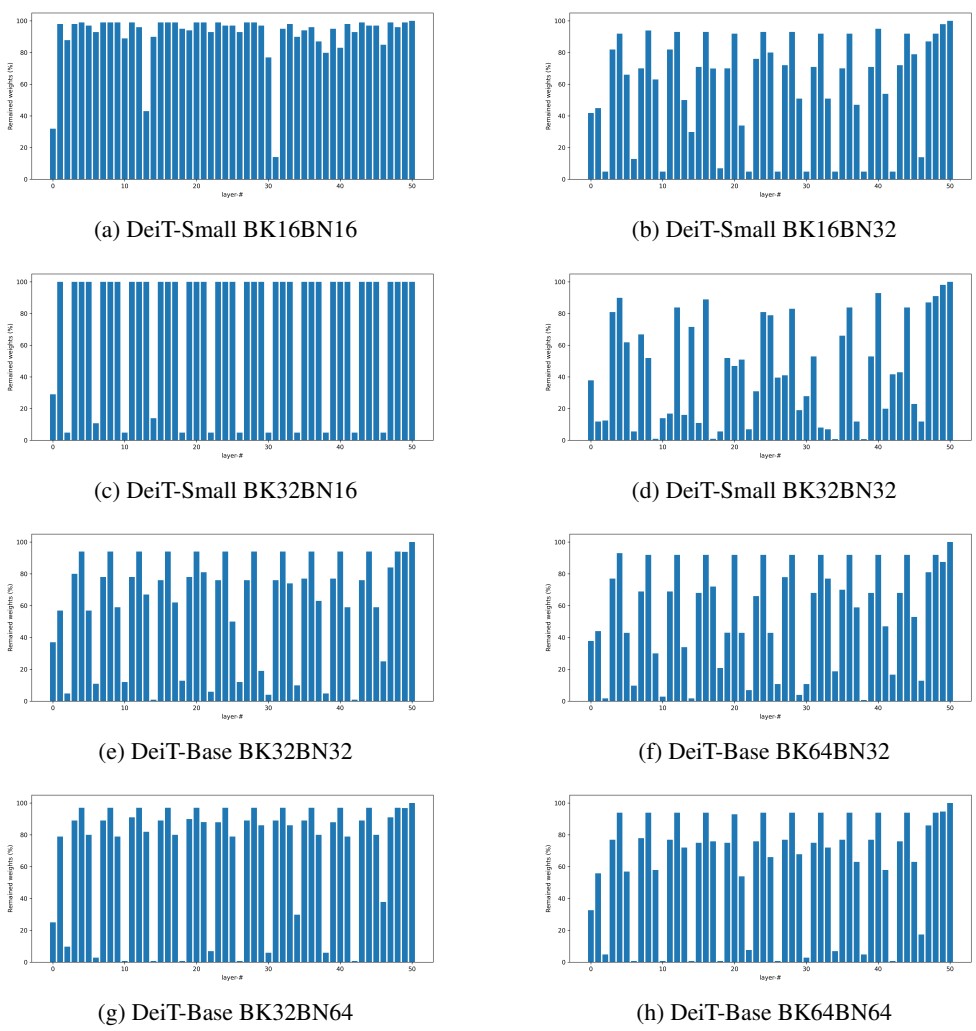

(a) DeiT-Small BK16BN16

(b) DeiT-Small BK16BN32

(c) DeiT-Small BK32BN16

(d) DeiT-Small BK32BN32

(e) DeiT-Base BK32BN32

(f) DeiT-Base BK64BN32

(g) DeiT-Base BK32BN64

(h) DeiT-Base BK64BN64

Figure 3: Layerwise pruning rate allocation for 50 layers of DeiT-Small and DeiT-Base in different blocksize. The height of bars indicate the percentage of survived connections in the layer.

## B    VISUALIZATIONS

**Layerwise-sparsity Allocation.** As introduced, our method optimizes the layerwise-sparsity allocation given a global FLOPs target. We visualize the optimization results for each individual settings in Tab. 3 as below. we notice some interesting observations. First, LSP preserved more connections in the last transformer blocks including the classification head on both DeiT-B and DeiT-S on different pruning ratios, and the classification head is almost kept unpruned in all cases. This observation is intuitive where many prior compression works showed that the last layer is crucial to the performance. Second, on both DeiT-B and DeiT-S, we notice the projection layers after MHA in particular, which are $[3 + 4n]$-th bars in the figures ($n$ from 0 to 11) are always getting high pruning ratio, showing that projection layers in ViTs have more redundancies and effect the model performance the least. The above patterns are all automatically learned from our second-order pruning layer-wise sparsity allocation algorithm, showing the effectiveness of our method.

**Block-sparsity Visualizations.** We also visualize the block-sparsity pattern generated for several layer weights in Fig. 4. White regions are unpruned connections and black regions are pruned connections.

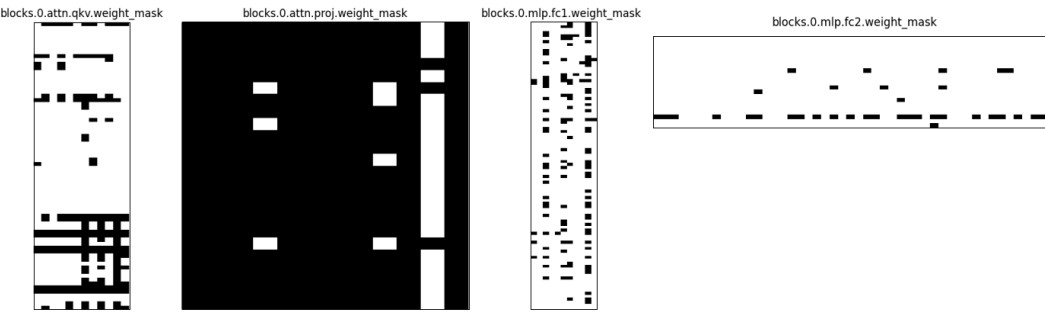

Figure 4: Block-sparse pattern generated for different ViT layers.

**Self-attention Maps.** As shown in Fig. 5, we followed the same self-attention maps visualization process adopted in Chen et al. (2021b); Cordonnier et al. (2019) to show potential influence of the pruning on the attention behaviors in transformer multi-head attention. We observe that our LSP block-sparsity pruning scheme displays a coarse and discretized pattern in a lot of attention heads across transformer blocks. This is due to the blocksparse pruning in the weight layers affects a whole block of region of calculation at once, which decreases the possible choices of output values in the consequent layers. We also observe some completely inactive attention heads, similar to the behaviour in previous structural pruning model SViTE Chen et al. (2021b), which facilitates further inference speedup and power saving by directly discard the following computations within the transformer block. Compared to structural pruning scheme, our semi-structured scheme allows middle states between blank attention and the delicate pattern in dense model, preserving more attention information which is curcial to the model accuracy. Another observation is that on LSP-DeiT-Base FLOPs 45% model (bottom-middle), the last two attention layers have no active attention heads. As a result, the entire blocks can be discarded in calculation, which could possibly bring the reported FLOPs reduction even more.

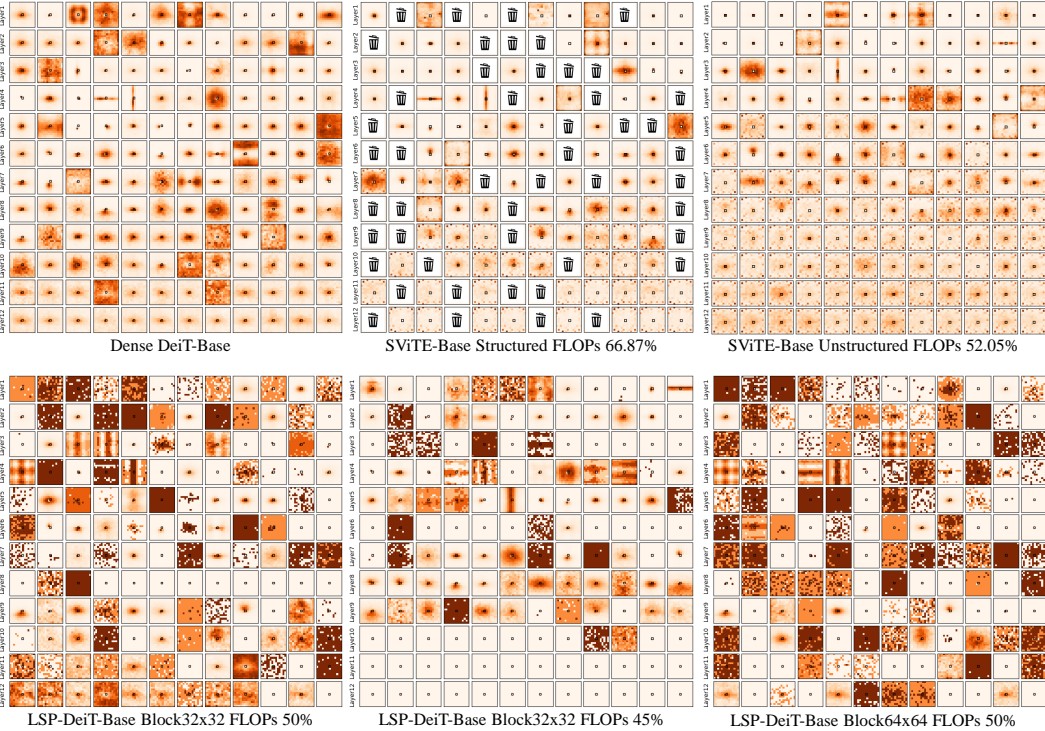

Figure 5: Attention probabilities for DeiT-Base, SViTE-Base Structured/Unstructured pruning models as well as our LSP-DeiT-Base model with 12 layers (rows) and 12 heads (columns) using visualization tools provided in Cordonnier et al. (2019).

## C  Hardware Benchmarking Results

We further evaluate the inference speedups and power efficiencies of our LSP pruning method on three types of hardware platforms: a RISC-V platform ViTA Chen et al. (2024), a DNN-targeted accelerator TPU V3 Jouppi et al. (2020), a GPU platform NVIDIA A100. Results show that our approach is able to bring noticeable improvements for ViT models on the hardware platforms, which demonstrate the effectiveness of our approach.

**ViTA Results.** ViTA Chen et al. (2024) is a novel DL acceleration platform based on RISC-V architecture with PEs (parallel exection) kernels supported in multiple blocksize configurations. Hence the speedup on ViTA can be achieved as closest to the theoretical target when the corresponding blocksized PE is selected for each LSP pruned model correctly. Our approach obtains $5.19\times$, $4.14\times$ and $1.62\times$ speedups for DeiT-Base, DeiT-Small and DeiT-Tiny on ViTA, respectively. Tab. 5 shows the details.

**TPU V3 Results.** We also adapt our block-sparse ViTs on high-performance DNN-targeted hardware accelerator Google TPU V3 Jouppi et al. (2020), adopting SCALE-sim Samajdar et al. (2020) to simulate the time cycle. Since TPU V3 only offers 1 type of MAC with block size fixed at $128 \times 128$, we expect less speedup than on RISC-V because all smaller-sized blocks from our tested configs (as large as $64 \times 64$) within $128 \times 128$ must be pruned to skip the computation, which means the hardware effective block-level sparsity is smaller than on VITA. Nevertheless, our approach still obtains noticeable speedups for the ViT models on TPU, bringing about at most $2.57\times$, $1.54\times$, $1.02\times$ speedups for DeiT-Base, DeiT-Small and DeiT-Tiny, respectively. Tab. 6 shows the results.

**A100 GPU Results.** we deploy on NVIDIA A100 40GB GPU with CUDA 11.8 and evaluate the end-to-end inference time and the runtime power consumption as shown in Tab, 7, where we also observe a power reduction up to 60.4% on DeiT-Base. Power consumption is measured by averaging `nvidia-smi`'s power meter over an adequate time period and subtracting the idle power consumption.

Table 5: Simulated Speedup on ViTA.

| Model | FLOPs Remained (%) | Blocksize | Inference Time (ms) | Speedup | Top-1 Accuracy |
|---|---|---|---|---|---|
| DeiT-Base | 100 | - | 16.49 | - | 81.8 |
|  | 25.8 | $32 \times 32$ | 4.19 | $3.93\times$ | 77.75 |
|  | 38.5 | $64 \times 32$ | 3.15 | $2.61\times$ | 78.1 |
|  | 19.8 | $32 \times 64$ | 1.59 | $\mathbf{5.19\times}$ | 61.42 |
|  | 65.9 | $64 \times 64$ | 2.71 | $1.52\times$ | 67.67 |
| DeiT-Small | 100 | - | 8.3 | - | 79.8 |
|  | 49.9 | $32 \times 32$ | 4.17 | $1.99\times$ | 73.32 |
|  | 24.5 | $32 \times 32$ | 2.01 | $\mathbf{4.14\times}$ | 65.12 |
|  | 91.4 | $32 \times 16$ | 7.68 | $1.08\times$ | 79.09 |
|  | 71.3 | $16 \times 32$ | 5.94 | $1.4\times$ | 78.2 |
| DeiT-Tiny | 100 | - | 4.24 | - | 72.2 |
|  | 61.2 | $16 \times 16$ | 2.61 | $\mathbf{1.62\times}$ | 69.06 |

## D  Transfer Learning to Downstream Tasks

To evaluate the generalizability of LSP on downstream tasks, We further evaluate on transferred learning performance of our method on the downstream Cityscapes Cordts et al. (2016) segmentation task. We first pretrained a DeiT-B/384 pruned by 50% FLOPS on imagenet for 27 epochs and further use it as a backbone in a recent segmentation model SETR Zheng et al. (2021) and train on Cityscapes dataset using the configuration of "`SETR_Naive_DeiT_768x768_80k_cityscapes_bs_8`". (Here "pretrain" refers to the same finetuning process in previous ImageNet experiments, to extinguish between finetuning in cityscapes dataset.) Tab. 8 compares the val mIoU of the pruned backbone and the original performance. We only observe a performance degration of mere 1.27 mIoU. A visualization of the qualitative performance of LSP pruned segmentation model is also demontrated in Fig. 6, showing that the pruned model retains great visual quality even remaining only 50% FLOPs.

Table 6: Simulated Speedup on TPU V3.

| Model | FLOPs Remained (%) | Blocksize | Inference Time (ms) | Speedup | Top-1 Accuracy |
|---|---|---|---|---|---|
| DeiT-Base | 100 | - | 18.08 | - | 81.8 |
| | 25.8 | $32 \times 32$ | 11.46 | $1.57\times$ | 77.75 |
| | 38.5 | $64 \times 32$ | 12.62 | $1.43\times$ | 78.1 |
| | 19.8 | $32 \times 64$ | 7.26 | $\mathbf{2.57}\times$ | 61.42 |
| | 65.9 | $64 \times 64$ | 7.59 | $2.46\times$ | 67.67 |
| DeiT-Small | 100 | - | 2.7 | - | 79.8 |
| | 49.9 | $32 \times 32$ | 2.13 | $1.27\times$ | 73.32 |
| | 24.5 | $32 \times 32$ | 1.75 | $\mathbf{1.54}\times$ | 65.12 |
| | 91.4 | $32 \times 16$ | 2.7 | $1\times$ | 79.09 |
| | 71.3 | $16 \times 32$ | 2.62 | $1.03\times$ | 78.2 |
| DeiT-Tiny | 100 | - | 0.00063 | - | 72.2 |
| | 61.2 | $16 \times 16$ | 0.00061 | $\mathbf{1.02}\times$ | 69.06 |

Table 7: Simulated Speedup and power consumptions on A100 GPU.

| Model | FLOPs Remained (%) | Blocksize | Batchsize | Inference Time (ms) | Speedup | Power (W) |
|---|---|---|---|---|---|---|
| DeiT-Base | 100 | - | 4 | 10.76 | - | 167 |
| | 25.8 | $32 \times 32$ | 4 | 6.16 | $1.75\times$ | 120(71.8%) |
| | 38.5 | $64 \times 32$ | 4 | 6 | $\mathbf{1.79}\times$ | 147(88%) |
| | 19.8 | $32 \times 64$ | 4 | 6.11 | $1.76\times$ | $\mathbf{101(60.4}\%)$ |
| | 65.9 | $64 \times 64$ | 4 | 6.5 | $1.66\times$ | 122(73%) |
| DeiT-Small | 100 | - | 16 | 11.57 | - | 195 |
| | 49.9 | $32 \times 32$ | 16 | 8.12 | $1.42\times$ | 160(82%) |
| | 24.5 | $32 \times 32$ | 16 | 6.02 | $\mathbf{1.92}\times$ | $\mathbf{142(72.8}\%)$ |
| | 91.4 | $32 \times 16$ | 16 | 10.8 | $1.07\times$ | 192(98%) |
| | 71.3 | $16 \times 32$ | 16 | 9.62 | $1.2\times$ | 181(92.8%) |
| DeiT-Tiny | 100 | - | 256 | 5.12 | - | 216 |
| | 61.2 | $16 \times 16$ | 256 | 4.15 | $\mathbf{1.23}\times$ | $\mathbf{207(95.8}\%)$ |

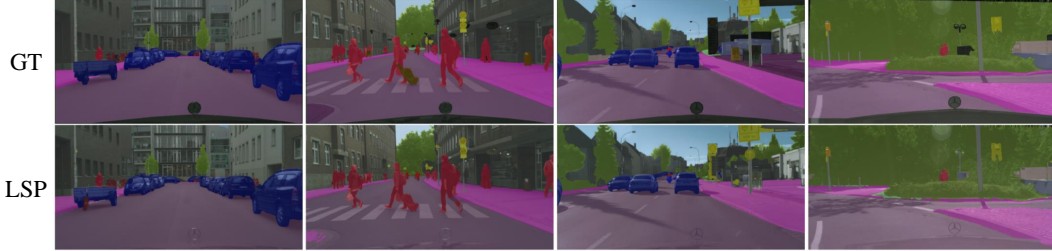

Figure 6: Qualitative result of segmentation masks on Cityscapes validation set predicted by SETR Zheng et al. (2021) with DeiT-Base/384 backbone remaining 50% FLOPs. First row shows ground truth mask and second row shows the predicted masks of pruned model.

Table 8: Segmentation results on Cityscapes valiadtion dataset.

| Backbone | Method | FLOPs (G) | mIoU |
|---|---|---|---|
| DeiT-Base/384 | Dense | 17.6 | 78.66 |
| DeiT-Base/384 | LSP | 8.8 | 77.39 |

