# OpenReview forum: "LSP: Low-Power Semi-structured Pruning for Vision Transformers"
_ICLR.cc/2024/Conference — Submitted to ICLR 2024_

### Official Review · Reviewer_yotU · 2023-10-31

**Soundness:** 2 fair
**Presentation:** 2 fair
**Contribution:** 2 fair
**Rating:** 5
**Confidence:** 4

**Summary:**

The paper proposes a block-structured pruning approach to prune the network, which leverages the block-wise structure of linear layers as a more efficient pruning scheme for ViTs compared to unstructured or channel-wise structured pruning. They propose a hardware-aware learning objective is introduced to optimize the block pruning scheme, which maximizes speedup and minimizes power consumption during inference. They use a lightweight algorithm utilizing second-order Taylor approximation and empirical optimization is provided for post-training pruning of ViTs.

**Strengths:**

1. The semi-structured (block-wise) pruning is a promising direction to explore.
2. The paper is easy to follow.
3. ViT compression is a important topic as the requirement in both industrial applications and academic research.

**Weaknesses:**

1. The introduction and discussion on network pruning appear to omit several milestone works in the field. It seems the authors may not thoroughly discuss the milestone contributions in the related works. There's a potential confusion between pruning at initialization and post-training pruning. Notably, pivotal works such as Han 2015 [1, 2] and HRank 2020 [3] are absent, while references to some arXiv papers are included. I recommend that the authors structure the related works section with more precise logic. Also, the authors claim that CNNs pruning can be divided into three categories, yet only two are presented.
2. What is the i.d.d. in Assumption 1?
3. The experimental results presented seem limited. In Table 1, the authors only explore DeiT-small and Base. It would be beneficial if the authors expanded their experiments to include, at least, DeiT-Tiny, T2T, and other ViTs, considering various pruning ratios. Also, the improvement from UVC is minimal on Deit-Base model.

[1] Deep Compression: Compressing Deep Neural Networks with Pruning, Trained Quantization and Huffman Coding.
[2] Pruning Convolutional Neural Networks for Resource Efficient Inference
[3] HRank: Filter Pruning using High-Rank Feature Map

**Questions:**

Please check Weakness 1 and 3.

---

> ### Author Response · Authors · 2023-11-22
> **(Q1 Q2) Thanks for your great questions. Here are the answers to Question 1, 2.**
>
> ## Q1
> The introduction and discussion on network pruning appear to omit several milestone works in the field. It seems the authors may not thoroughly discuss the milestone contributions in the related works. There's a potential confusion between pruning at initialization and post-training pruning. Notably, pivotal works such as Han 2015 [1, 2] and HRank 2020 [3] are absent, while references to some arXiv papers are included. I recommend that the authors structure the related works section with more precise logic. Also, the authors claim that CNNs pruning can be divided into three categories, yet only two are presented.
>
> ## Answer to Q1
> We appreciate your advice to improve the related works and we have substantially revised the related work. Regarding the confusion between pruning at initialization and post-training pruning, we wish to clarify here that the difference between this two scheme is that pruning at initialization determines the pruning mask from randomly initialized network weights while post-training pruning obtain pruning mask from converged model. We also re-organized related works into 3 categories: unstructured pruning, semi-structured pruning, structured (channel/filter-wise) pruning to better fit the main theme in this paper. For each category, we summarized their main features and discussed the representative works including the pivotal works. We added the mentioned references and other important citations which were absent. Please also feel free to let us know if other references are missed. The revised related work section can be found in our latest uploaded manuscript.
>
> ---
>
> ## Q2
> What is the i.d.d. in Assumption 1?
>
> ## Answer to Q2
> The term has been corrected as “i.i.d.” in the revised draft, referring to “Independent and identically distributed”. Sorry for this typo.

---

> ### Author Response · Authors · 2023-11-22
> **(Q3) Thanks for your great questions. Here are the answers to Question 3.**
>
> ## Q3
> The experimental results presented seem limited. In Table 1, the authors only explore DeiT-small and Base. It would be beneficial if the authors expanded their experiments to include, at least, DeiT-Tiny, T2T, and other ViTs, considering various pruning ratios. Also, the improvement from UVC is minimal on Deit-Base model.
>
> ## Answer to Q3
> We added imagenet experiments on another three ViT models: Deit-T, Swin-B and Swin-T. The results are still promising on those ViT variants. On Deit-T, we obtain around 50% pruning ratio regarding the FLOPs with a 3.14% accuracy drop. On Swin-B, we obtain a 50% pruning ratio with 3.9% accuracy loss. On Swin-T, we get only a 1.2% accuracy drop from dense model with 77% FLOPs remained, and a 1.96% accuracy drop with 71.1% FLOPs remained. We compared with X-Pruner [4] on Swin-T which also reported results on the model. With 71.1% FLOPs remained, our method outperformed X-Pruner [4] by 0.7%. We noticed that another paper SPViT [5] also reported results on Swin Transformers. We didn’t include SPViT [5] in the table because their method focuses on tokens pruning but our method is about weights pruning, so it is not an apple-to-apple comparison.
>
> > ### Table 1. Results on other ViT models
> | Model  | Method| Sparsity (%) | FLOPs Remained (%) |  Blocksize | Top-1 | Top-1 diff |
> |:------:|:-----:|:------------:|:-----:|:-----:|:-----:|:-----:|
> | DeiT-T 	| Dense | 100   | 100   | - | 72.2  | - |
> | DeiT-T 	| LSP | 61.2   | 50   | 16x16 | 69.06 | -3.14 |
> | Swin-B  	| Dense | 100  | 100  | - | 83.5 | - |
> | Swin-B  	| LSP| 57.6 | 50 | 32x32  | 79.6  | -3.9 |
> | Swin-T  	| Dense| 100 | 100  | - |  81.2 | - |
> | Swin-T | X-Pruner [4] | - | 71.1 |  - |  78.55 | -2.65 |
> | Swin-T  	| LSP | 70 | 71.1  | 16x16  | 79.24  | -1.96 |
> | Swin-T  	| LSP | 79 | 77  | 16x16 | 80.0  | -1.2 |
>
>
>
> Regarding the result of DeiT-B, the improvement from UVC is indeed less prominent than that on DeiT-S (0.24% vs 2.0%). However, we want to point out that our comparisons with UVC are actually not fair, because UVC prunes both tokens and weights, but our method prunes weights only as we focus on developing a block-structure pruning scheme for weights in the paper. It is also worth mentioning that our approach with only weights pruned still outperforms UVC with both tokens and weights pruned. We are running the results of our approach with also tokens pruned and we believe the results can be better after incorporating tokens pruning into our approach. We will update and add them in the manuscript once done. Thanks for this valuable insight.
>
>
> Regards,
>
> Authors
>
> ---
>
> ### References
>
> [1] Han, Song, Huizi Mao, and William J. Dally. "Deep compression: Compressing deep neural networks with pruning, trained quantization and huffman coding." arXiv preprint arXiv:1510.00149 (2015).
>
> [2] Molchanov, Pavlo, et al. "Pruning Convolutional Neural Networks for Resource Efficient Inference." International Conference on Learning Representations. 2016.
>
> [3] Lin, Mingbao, et al. "Hrank: Filter pruning using high-rank feature map." Proceedings of the IEEE/CVF conference on computer vision and pattern recognition. 2020.
>
> [4] Yu, Lu, and Wei Xiang. "X-Pruner: eXplainable Pruning for Vision Transformers." Proceedings of the IEEE/CVF Conference on Computer Vision and Pattern Recognition. 2023.
>
> [5] Kong, Zhenglun, et al. "Spvit: Enabling faster vision transformers via latency-aware soft token pruning." European Conference on Computer Vision. Cham: Springer Nature Switzerland, 2022.

---

### Official Review · Reviewer_d35y · 2023-10-31

**Soundness:** 3 good
**Presentation:** 3 good
**Contribution:** 3 good
**Rating:** 5
**Confidence:** 5

**Summary:**

This paper proposes a new block-structured sparse method, which is more fine-grained than the channel-wise structured sparse methods and can also achieve speedup due to the structured sparse pattern. And this method also introduces a novel hardware-aware learning object to reduce the power consumption during inference.

**Strengths:**

* The proposed block-structured sparse method can be a good trade-off for the model accuracy and the FLOPs reduction.

**Weaknesses:**

* This method is only applied to the linear layers in vision transformers. However, the multi-head attentions in the model also take a large proportion of the FLOPs, especially for the inputs with large sizes.
* The power consumption only considers the matrix multiplication operations, which is inaccurate because the memory accesses take a large proportion of energy consumption. The performance of vision transformers is usually bounded by the memory accesses, especially for those edge devices. So memory accesses should also be taken into account.
* The experiments are only conducted on the DeiT models, which use global attention. It is better to conduct more experiments on other vision transformer models, such as the Swin Transformer, which has been widely used nowadays.
* Experiments are insufficient. The authors only evaluate the proposed method on the classification task. It is better to discuss the generalization of the method and evaluate different tasks.
* Only show the FLOPs reduction, without the real speedup. Even with the adoption of block-wise sparsity, there is still some overhead during inference, especially for the sparse model with small block shapes. If the paper can show the real speedup, it will be more convincing.

**Questions:**

* As for the estimated power consumption, i.e., eq. 7, when used this to constrain the power consumption, as shown in eq. 8, what’s the difference with the direct constraint on the FLOPs? It seems like an additional parameter $p_m$ has been added to the FLOPs constraint.
* Were all the experiments fine-tuned for 300 epochs? If that is the case, it should not be referred to as a post-training method, as illustrated in the abstract. In addition, it is unfair to compare the baseline in Table 1. The baseline in Table 1 is not training with the knowledge distillation, but the LSP models are fine-tuned with 300 epochs with knowledge distillation. And the accuracy of the distilled DeiT-Small and DeiT-Base should be 81.2% and 83.4%, respectively.

---

> ### Author Response · Authors · 2023-11-22
> **(Q1 Q2 Q3) Thanks for your great questions. Here are the answers to Question 1, 2, 3.**
>
> ## Q1
> This method is only applied to the linear layers in vision transformers. However, the multi-head attentions in the model also take a large proportion of the FLOPs, especially for the inputs with large sizes.
>
> ---
>
> ## Answer to Q1
> We agree with reviewer d35y that multi-head attentions take a large portion of FLOPs especially in cases with high-resolution input images. Pruning multi-head attentions and pruning weights are usually considered as two different directions. Like the previous ViT pruning works [1,2,3], our work focuses on weights pruning. On the other hand, the two pruning schemes actually compensate with each other — by combining both multi-head attention pruning and weights pruning, one can achieve higher speedups and more power reductions. In our current setup, image size is fixed to 224x224 on the ImageNet dataset, which is the same as other baselines [1,2,3]. The proportion of FLOPs of multi-head attention under our current setting is about 4% on ViTs and DeiTs, respectively, which is relatively small. As pointed out by d35y, for high-resolution image tasks (e.g., image understanding, image segmentation, etc.), input size can be large and multi-head attentions take a large proportion of FLOPs, making multi-head attention pruning necessary. In such cases, it is a good idea to incorporate multi-head attention pruning and combine both multi-head attention pruning and weight pruning in a joint framework to maximize the computation reduction. We would like to consider this direction as our future work for further study and research. Thanks for this insight.
>
> ---
>
> ## Q2
> The power consumption only considers the matrix multiplication operations, which is inaccurate because the memory accesses take a large proportion of energy consumption. The performance of vision transformers is usually bounded by the memory accesses, especially for those edge devices. So memory accesses should also be taken into account.
>
> ## Answer to Q2
> We agree with d35y that both computation and memory cost energy. Pruning weights also helps to reduce the energy cost of memory, since pruning makes model size smaller leading to less data read/write from off-chip memory to on-chip memory. More than that, we incorporate hardware-aware constraints into our learning framework so that given a target pruning ratio our approach lowers the computation as much as possible. As a result, our approach reduces the energy cost for both memory and computation. We added results on hardware platforms to further demonstrate the effectiveness of our approach and observed a noticeable power reduction up to 40% on GPU. We also conducted experiments on DNN-targeted accelerators designed for mobile/edge scenarios (a RISC-V platform ViTA[4] and Google TPU v3 [5]). Please check the detailed results in our answers to Question 5. Thanks a lot for this important question.
>
> ---
>
> ## Q3
> The experiments are only conducted on the DeiT models, which use global attention. It is better to conduct more experiments on other vision transformer models, such as the Swin Transformer, which has been widely used nowadays.
>
> ## Answer to Q3
> We added experiments on the Swin Transformer, including Swin-B and Swin-T. Our results on the Swin Transformer are still promising. Specifically, on Swin-B, with 50% FLOPs remained, the accuracy loss is 3.9%. On Swin-T, with 71.1% FLOPs remained, the accuracy loss is 1.96%. We compared with X-Pruner[6] which also reported results on the Swin Transformer. On Swin-T, our approach outperforms X-Pruner by 0.7% when FLOPs reduction is around 29%. Table 1 below shows the details. We are still running results at other more settings for the Swin Transformer. We will update and add them in the revised manuscript once done. Please feel free to let us know if results on other variants or models are needed. Thanks for this good suggestion.
> > ### Table 1. Swin Transformer results.
> | Model  | Method| Sparsity (%) | FLOPs Remained (%) |  Blocksize | Top-1 | Top-1 diff |
> |:------:|:-----:|:------------:|:-----:|:-----:|:-----:|:-----:|
> | Swin-B  	| Dense | 100  | 100  | - | 83.5 | - |
> | Swin-B  	| LSP| 57.6 | 50 | 32x32  | 79.6  | -3.9 |
> | Swin-T  	| Dense| 100 | 100  | - |  81.2 | - |
> | Swin-T | X-Pruner [6] | - | 71.1 | - |  78.55 | -2.65 |
> | Swin-T  	| LSP | 70 | 71.1  | 16x16  | 79.24  | -1.96 |
> | Swin-T  	| LSP | 79 | 77  | 16x16 | 80.0  | -1.2 |

---

> ### Author Response · Authors · 2023-11-22
> **(Q4 Q5)(Part 1/2) Thanks for your great questions. Here are the answers to Question 4, 5. (cont'd)**
>
> ## Q4
> _<Experiments are insufficient. Should evaluate different tasks>_
>
> ## Answer to Q4
> We added additional experiments on other tasks: image segmentation and image retrieval. On image segmentation task, we evaluated the results on Cityscapes dataset. We pretrained a DeiT-B/384 model pruned by 50% FLOPS and used it as a backbone in a SOTA segmentation architecture SETR[7]. Results show that our approach can prune the FLOPs by 50% with only 1.27% accuracy loss. A visualization of the prediction of the pruned segmentation model is also included in the appendix of revised manuscript (see Fig. 6).
>
> > ### Table 2. Transferred segmentation results on Cityscapes dataset
> | Backbone | Dataset | Method | FLOPs (G) | FLOPs remained (%) | mIoU | mIoU loss |
> |:------:|:-----:|:------------:|:-----:|:-----:| :-----:| :-----:|
> | DeiT-B/384 	| Cityscapes | Dense | 17.6 | 100 | 78.66 | - |
> | DeiT-B/384 	| Cityscapes | LSP | 8.8| 50 | 77.39 | -1.27 |
>
> **We further evaluate on a downstream image retrieval task using the Oxford building dataset. As shown in Table 7, We achieve lossless retrieval on DeiT-S with FLOPs 49% remaining.**
>
> > ### Table 7. Image Retrieval task on Oxford Building dataset
> | Backbone | Method | FLOPs remained (%) | Top-1 |
> |:------:|:-----:|:------------:|:-----:|
> | DeiT-S 	| Dense | 100| 65.4 |
> | DeiT-S 	| LSP | 49 | 65.4 |
>
>
> ---
>
> ## Q5
> _<Need real speedup statistics>_
>
> ## Answer to Q5
> We added experiments on three real hardware platforms: a RISC-V platform ViTA [4], a DNN-targeted accelerator TPU v3 [5], a GPU platform NVIDIA A100. Results show that our approach brings noticeable improvements for the real speedup on these hardware platforms, which demonstrate the effectiveness of our approach.
>
> The first group of evaluation is on a RISC-V platform ViTA [4]. Our approach brings about 5.19x, 4.14x and 1.62x speedups for DeiT-B, DeiT-S and DeiT-T on ViTA, respectively, without hurting the accuracy. Table 3 below shows the detailed results.
>
> > ### Table 3. Hardware Speedup on VITA.
> | Model | FLOPs Remained (%) | Blocksize | Inference Time (ms) | Speedup | Top-1 Accuracy |
> |:------:|:-----:|:-----:|:-----:|:-----:|:-----:|
> | DeiT-B 	| 100 | -   | 16.49  | - | 81.8 |
> | DeiT-B 	| 25.8 | 32x32  | 4.19 | 3.93x | 77.75 |
> | DeiT-B 	| 38.5 | 64x32   |3.15  | 2.61x |  78.1 |
> | DeiT-B 	| 19.8 | 32x64   | 1.59 | 5.19x | 61.42 |
> | DeiT-B 	| 65.9 | 64x64   | 2.71 |  1.52x | 67.67 |
> | DeiT-S 	| 100 | -  | 8.3 |  - | 79.8 |
> | DeiT-S 	| 49.9 | 32x32  | 4.17 | 1.99x | 73.32 |
> | DeiT-S 	| 24.5 | 32x32  | 2.01 |  4.14x | 65.12 |
> | DeiT-S 	| 91.4 | 32x16  | 7.68 |  1.08x | 79.09  |
> | DeiT-S 	| 71.3 | 16x32  | 5.94 |  1.4x | 78.2  |
> | DeiT-T 	| 100 | -  | 4.24 |  - | 72.2 |
> | DeiT-T 	| 61.2 | 16x16  | 2.61 | 1.62x | 69.06 |
>
>
> In the second group, we adapt our block-sparse ViTs on high-performance DNN-targeted hardware accelerator Google TPU V3 [5], adopting SCALE-sim [8] to simulate the time cycle. Our approach also obtains noticeable speedups on TPU, bringing about at most 2.57x, 1.54x, 1.02x speedups for DeiT-B, DeiT-S and DeiT-T, respectively. Table 4 below shows the results.
>
> We do observe less speedups for smaller blocksizes, but speedups are still noticeable, because the overhead in noncalculation operations, such as memory IO (900GB/s memory bandwidth on TPU v3), is highly efficient on those platforms which are equipped with well-designed data-transfer overlapping.  When the FLOPs are reduced more, speedup is even higher (e.g., DeiT-Base with blocksize 64x64 remaining FLOPs of 65.9% achieves 5.19x and 2.57x speedups on VITA and TPU respectively).
>
>
> > ### Table 4. Hardware Speedup on TPU V3.
> | Model  | FLOPs Remained (%) | Blocksize | Inference Time (s) | Speedup | Top-1 Accuracy  |
> |:------:|:------------------:|:---------:|:------------------:|:-------:|:---------------:|
> | DeiT-B | 100                | -         | 18.08              | -       | 81.8            |
> | DeiT-B | 25.8               | 32x32     | 11.46              | 1.57x   | 77.75           |
> | DeiT-B | 38.5               | 64x32     | 12.62              | 1.43x   | 78.1            |
> | DeiT-B | 19.8               | 32x64     | 7.26               | 2.57x   | 61.42           |
> | DeiT-B | 65.9               | 64x64     | 7.59               | 2.46x   | 67.67           |
> | DeiT-S | 100                | -         | 2.7                | -       | 79.8            |
> | DeiT-S | 49.9               | 32x32     | 2.13               | 1.27x   | 73.32           |
> | DeiT-S | 24.5               | 32x32     | 1.75               | 1.54x   | 65.12           |
> | DeiT-S | 91.4               | 32x16     | 2.7                | 1x      | 79.09           |
> | DeiT-S | 71.3               | 16x32     | 2.62               | 1.03x   | 78.2            |
> | DeiT-T | 100                | -         | 0.00063            | -       | 72.2            |
> | DeiT-T | 61.2               | 16x16     | 0.00061            | 1.02x   | 69.06           |

---

> ### Author Response · Authors · 2023-11-22
> **(Q4 Q5)(Part 2/2) Thanks for your great questions. Here are the answers to Question 4, 5. (cont'd)**
>
> In the third group, we deploy on NVIDIA A100 40GB GPU with CUDA 11.8 and evaluate the end-to-end inference time which is shown in Table 5. We also observed noticeable speedups by up to 1.79x, 1.92x, and 1.23x on DeiT-B, DeiT-S, and Deit-T, respectively.
>
> > ### Table 5. Hardware Speedup and power consumption on A100 GPU.
> | Model  | FLOPs Remained (%) | Blocksize | Batchsize | Inference Time (ms) | Speedup | Power (W) | Top-1 Accuracy  |
> |:------:|:------------------:|:---------:|:---------:|:-------------------:|:-------:|:---------:|:---------------:|
> | DeiT-B | 100                | -         | 4         | 10.76               | -       | 167       | 81.8            |
> | DeiT-B | 25.8               | 32x32     | 4         | 6.16                | 1.75    | 120 (71.8%)       | 77.75           |
> | DeiT-B | 38.5               | 64x32     | 4         | 6                   | 1.79    | 147 (88%)       | 78.1            |
> | DeiT-B | 19.8               | 32x64     | 4         | 6.11                | 1.76    | 101 (60.4%)      | 61.42           |
> | DeiT-B | 65.9               | 64x64     | 4         | 6.5                 | 1.66    | 122 (73%)      | 67.67           |
> | DeiT-S | 100                | -         | 16        | 11.57               | -       | 195       | 79.8            |
> | DeiT-S | 49.9               | 32x32     | 16        | 8.12                | 1.42    | 160 (82%)      | 73.32           |
> | DeiT-S | 24.5               | 32x32     | 16        | 6.02                | 1.92    | 142 (72.8%)      | 65.12           |
> | DeiT-S | 91.4               | 32x16     | 16        | 10.8                | 1.07    | 192 (98%)      | 79.09           |
> | DeiT-S | 71.3               | 16x32     | 16        | 9.62                | 1.2     | 181 (92.8%)      | 78.2            |
> | DeiT-T | 100                | -         | 256       | 5.12                | -       | 216       | 72.2            |
> | DeiT-T | 61.2               | 16x16     | 256       | 4.15                | 1.23    | 207 (95.8%)      | 69.06           |
>
> More discussion about the hardware results and the detailed setup information of the experiences can be found in our response to [cWB3’s Question 1](https://openreview.net/forum?id=FoqZKsH9sE&noteId=reicKXpTyq). We have also modified the manuscript accordingly. Thanks a lot for this important suggestion.

---

> ### Author Response · Authors · 2023-11-22
> **(Q6 Q7) Thanks for your great questions. Here are the answers to Question 6, 7.**
>
> ## Q6
> As for the estimated power consumption, i.e., eq. 7, when used this to constrain the power consumption, as shown in eq. 8, what’s the difference with the direct constraint on the FLOPs? It seems like an additional parameter pm has been added to the FLOPs constraint.
>
> ## Answer to Q6
> The difference is that in constraint Eq. 8, we have the parameter $p_m$ (the unit power cost of an individual within-block matmul), which does not exist in the FLOPs calculation. Note that the $p_m$ is essential to calculate the power consumption. In empirical inference, the power consumption of each unit calculation in PEs is subject to complicated internal and external influence where the most accurate way is to construct a lookup table for the $p_m$ for different layers in the transformer models.
>
> ---
>
> ## Q7
> Were all the experiments fine-tuned for 300 epochs? If that is the case, it should not be referred to as a post-training method, as illustrated in the abstract. In addition, it is unfair to compare the baseline in Table 1. The baseline in Table 1 is not training with the knowledge distillation, but the LSP models are fine-tuned with 300 epochs with knowledge distillation. And the accuracy of the distilled DeiT-Small and DeiT-Base should be 81.2% and 83.4%, respectively.
>
> ## Answer to Q7
> Post-train-pruning is defined as the pruning methods that determine the pruning masks from converged pretrain models [9], which is the case in our method. Most existing post-train-pruning methods need to go through post-training (or called fine-tuning), which may even need to be iterated. Post-training is crucial to restoring the accuracy loss due to removing weights and is also used by prior ViT compression works such as UVC [1], VTP [2] and SPViTE [3], etc. As a result, we select the number of fine-tuning epochs of the pruning model as 300 to make sure the model performance is well restored, but actually we observe that the models converge well and already get competitive accuracy after 100-150 epochs. Another aspect is that, since we determine the pruning mask from converged pretrain models, the finetuning can be performed with sparse training libraries, which costs much less than a full dense training where both forward and backward can be accelerated. This also draws a major difference between a finetuning in post-training pruning and regular training.
>
> Regarding the knowledge distillation, the most competitive baseline UVC [1] does incorporate knowledge distillation in their experiments. They mentioned choosing the pretrained uncompressed model as the teacher model.
>
> Regarding the dense ViT performance, we reported the dense DeiT-Base/DeiT-Small top-1 as 81.8 and 79.2 respectively without distillation, which is reasonable for the reasons as follows. We evaluate all our method using the output classification token only, which is aligned with our baseline UVC[3]. In contrast, Distilled DeiTs Top-1 are obtained by averaging both the output classification and distillation tokens. Our baseline UVC[1] also listed the accuracies without distillation of dense DeiT-B and DeiT-S in their comparisons. So we keep the same comparison method to compare the accuracy drop at the same level.
>
> ---
>
> Hope this addresses your concerns and we are open to further discussions.
>
> Regards,
>
> Authors
>
> ### References
>
> [1] Yu, Shixing, et al. "Unified Visual Transformer Compression." International Conference on Learning Representations. 2021.
>
> [2]Zhu M, Tang Y, Han K. Vision transformer pruning[J]. arXiv preprint arXiv:2104.08500, 2021.
>
> [3] Chen, Tianlong, et al. "Chasing sparsity in vision transformers: An end-to-end exploration." Advances in Neural Information
> Processing Systems 34 (2021): 19974-19988.
>
> [4] C.Y. Chen et al., “ViTA: A Highly Efficient Dataflow and Architecture for Vision Transformers,” 2024 DATE.
>
> [5] Jouppi, Norman P., et al. "A domain-specific supercomputer for training deep neural networks." Communications of the ACM 63.7 (2020): 67-78.
>
> [6] Yu, Lu, and Wei Xiang. "X-Pruner: eXplainable Pruning for Vision Transformers." Proceedings of the IEEE/CVF Conference on Computer Vision and Pattern Recognition. 2023.
>
> [7] Zheng, Sixiao, et al. "Rethinking semantic segmentation from a sequence-to-sequence perspective with transformers." Proceedings of the IEEE/CVF conference on computer vision and pattern recognition. 2021.
>
> [8]  A. Samajdar, J. M. Joseph, Y. Zhu, P. Whatmough, M. Marina, and T. Krishna, “A Systematic Methodology for Characterizing Scalability of DNN Accelerators using SCALE-Sim,” in 2020 IEEE International Symposium on Performance Analysis of Systems and Software (ISPASS), 2020, pp. 58–68.
>
> [9] Wang, Huan et al. “Recent Advances on Neural Network Pruning at Initialization.” International Joint Conference on Artificial Intelligence (2021).

---

> > ### Comment · Reviewer_d35y · 2023-11-23
> >
> > Thanks for your detailed feedback. For the accuracy issue, there is a large accuracy drop, e.g. -3.9% in Swin-B. I think these results can not demonstrate the effectiveness of this method. The results of DeiT look fine, but I am not sure if this is due to the distillation since the comparison with the baseline dense models without distillation is a bit unfair. So I will keep my rating. I think it would be more convincing to have more ablation studies to discuss the effect of the distillation.

---

> > > ### Author Response · Authors · 2023-11-23
> > > **Thanks for your reply!**
> > >
> > > Thanks for your reply. Regarding the accuracy drop on Swin-B, we notice that current SOTA ViTs pruning papers who reported Swin transformer results like X-Pruner and SPViT only reduce the FLOPs by no more than 30%, and they do not provide Swin-B results either. While these methods rely on other sparsities on e.g. token, our method solely rely on weight pruning and managed to get Swin-B results with FLOPs 50% remaining, which is a technical breakthrough.
> > >
> > > On Swin-T, with comparison on the same level (FLOPs 71.1% remaining for both us and the baselines), we outperform to those token pruning or hybrid methods.
> > >
> > > As for the distillation, we have done an ablation study to remove distillation on DeiT-B, with sparsity of 50% for both w/ and w/o distillation. The results show that removing the distillation from training causes 0.4% accuracy drop, considered as marginal. The results are listed in the Table. 6. We will add the discussion in the revised manuscript.
> > >
> > > > ### Table 6. Ablation studies on Distillation
> > > | Model | Method| FLOPs Remained (%) | blocksize | Top-1 |
> > > |:------------:|:-----:|:-----:|:-----:|:-----:|
> > > | DeiT-B | w/ distill | 49 | 32x32 | 80.44 |
> > > | DeiT-B | w/o distill | 50 | 32x32 | 80.01 |
> > >
> > > Hope this clarifies your concerns.
> > >
> > > Best,
> > >
> > > Authors

---

### Official Review · Reviewer_rMt3 · 2023-10-31

**Soundness:** 3 good
**Presentation:** 3 good
**Contribution:** 3 good
**Rating:** 6
**Confidence:** 4

**Summary:**

This paper introduces a novel ViTs weight pruning algorithm dedicated to cut off energy consumption of inference by taking advantage of the characteristics of the ViT architecture, which mostly constructed by linear layers and proposed to use semi-structured (block-sparsity) pruning scheme to bridge the gap between fine-tuning stability and hardware friendliness.

**Strengths:**

1.	This paper introduces an optimal hardware-aware pruning objective designed for ViTs models within the block-structured pruning scheme.

2.	The proposed framework offers an efficient solution for the hardware-aware objective function through the use of a second-order Taylor approximation. It also introduces an empirical optimization method with a linear time complexity, providing a practical and effective approach.

**Weaknesses:**

1.	It seems like the block size is the same throughout the whole pruning process. If it is possible that the block size can be different at each layer, this may provide a more flexible pattern.

2.	At a certain Flops constraint, the layer-wise sparsity is computed by the proposed method. I would like to know the accuracy of the proposed method versus the traditional uniform approach for the same flops budget.

3.	How does the sparsity distribution vary according to different layers? Does the distribution show similar pattern among different models and Flops budgets?

4.	How is the performance of the Swin transformer?

5.	It would be great to show some visual results (e.g., attention maps) to prove the effectiveness when compared with other methods.

**Questions:**

Please refer to weaknesses.

---

> ### Author Response · Authors · 2023-11-22
> **(Q1 Q2 Q3) Thanks for your kind recognition and your questions! Here is the answer to Question 1,2,3**
>
> We sincerely appreciate the reviewer for recognition of our motivation, work's contributions and experimental effectiveness.
>
> We would also like to carefully respond to the questions as follows.
>
> ## Q1
> It seems like the block size is the same throughout the whole pruning process. If it is possible that the block size can be different at each layer, this may provide a more flexible pattern.
>
> ## Answer to Q1
> We agree with Reviewer rMt3 that it is possible to design different blocksizes for different layers and probably beneficial to the accuracy. However, to achieve the desired speedup and power reduction, hardware must also support various blocksize configurations. This could increase the difficulties and costs for hardware design. For example, if we deploy such flexible blocksize models directly on unified MACs hardware platform, such as TPU v3, the actual speedup would depend on the size of MAC on the platform (e.g. for a 64x64 blocksparse layer, to skip one area of 128x128 MAC calculation on TPU, all 4 64x64 blocks inside must be zero, which is far from likely). Such drop in speedup is also observed in our response to [this review](https://openreview.net/forum?id=FoqZKsH9sE&noteId=reicKXpTyq). On the other hand, if the hardware blocksize is relatively small, the accuracy and speedup of the models with flexible blocksizes is in fact equivalent to simply using the smallest blocksize everywhere. In conclusion, additional efforts in hardware design are necessary to enjoy the merit of such flexible scheme. We believe this topic is valuable and deserves more study among the community. We will also add related discussion regarding the possible flexible blocksize scheme in our manuscript. Thanks for this insight.
>
> ---
>
> ## Q2
> At a certain Flops constraint, the layer-wise sparsity is computed by the proposed method. I would like to know the accuracy of the proposed method versus the traditional uniform approach for the same flops budget.
>
> ## Answer to Q2
> Thanks for this suggestion. We added results of comparing with uniform pruning at the same FLOPs, and observed noticeable accuracy discrepancy between our proposed layerwise sparsity scheme and traditional uniform scheme. Specifically, on the DeiT-S model, with 50% FLOPs remained, our approach outperforms uniform pruning by 3.52%. On DeiT-T with 50% FLOPs remained, LSP also outperforms uniform pruning by 4.74%. This verifies that allocating layerwise sparsity according to the sensitivities of layers towards accuracy is able to boost the accuracy. Detailed results are shown in Table 1.
>
> > ### Table 1. Comparison with Uniform pruning.
> | Model  | Method | Sparsity (%) | FLOPs Remained (%) | Blocksize |Top-1 | Dense | Top-1 diff |
> |:------:|:-----:|:------------:|:-----:|:-----:|:-----:|:-----:|:-----:|
> | DeiT-S 	| LSP(Ours) | 92.2   | 50   | 16x16 | 80.86 | 79.8 | +0.89 |
> | DeiT-S 	| Uniform | 50 | 50 | 16x16 | 77.17  |  79.8 | -2.63 |
> | DeiT-T 	| LSP(Ours) | 61.2   | 50   | 16x16 | 69.06 | 72.2 | -3.14 |
> | DeiT-T 	| Uniform | 50 | 50 | 16x16 | 64.52  | 72.2 | -7.88 |
>
> ---
>
> ## Q3
> How does the sparsity distribution vary according to different layers? Does the distribution show similar pattern among different models and Flops budgets?
>
> ## Answer to Q3
> We plotted out the layerwise sparsity result for various settings and noticed some interesting observations. First, LSP preserved more connections in the classification head on both DeiT-B and DeiT-S on different pruning ratios, and the classification head is almost kept unpruned in all cases. This observation is intuitive where many prior compression works showed that the last layer is crucial to the performance. Second, on both DeiT-B and DeiT-S, we notice the projection layers after MHA are always getting high pruning ratio, showing that projection layers in ViTs have more redundancies and affect the model performance the least. The above patterns are all automatically learned from our second-order pruning layer-wise sparsity allocation algorithm, showing the effectiveness of our method. We added the figures of layerwise sparsity distribution and also included a visualization of the real block-sparse pruning masks in the appendix of revised manuscript. Please find the details in Fig. 3,4 in the appendix of the revised manuscript. Thanks a lot for this good suggestion.

---

> ### Author Response · Authors · 2023-11-22
> **(Q4 Q5) Thanks for your kind recognition and your questions! Here is the answer to Question 4,5 (cont'd)**
>
> ## Q4
> How is the performance of the Swin transformer?
>
> ## Answer to Q4
> We added additional experiments on Swin-B and Swin-T. Results show that our approach still performs well on both of the models. On Swin-B, we are able to prune the number of FLOPs by 50% without hurting the accuracy (loss = 3.9%). On Swin-T, the accuracy loss is only 1.96% when 30% FLOPs are pruned. Table 2 below shows the details. We compared with X-Pruner[1] which also reported results on the Swin-T, where our approach outperforms X-Pruner by 2.7% when FLOPs reduction is around 28%.
>
> > ### Table 2. Swin Transformer results.
> | Model  | Method| Sparsity (%) | FLOPs Remained (%) |  Blocksize | Top-1 | Top-1 diff |
> |:------:|:-----:|:------------:|:-----:|:-----:|:-----:|:-----:|
> | Swin-B  	| Dense | 100  | 100  | - | 83.5 | - |
> | Swin-B  	| LSP| 57.6 | 50 | 32x32  | 79.6  | -3.9 |
> | Swin-T  	| Dense| 100 | 100  | - |  81.2 | - |
> | Swin-T | X-Pruner [1] | / | 71.1 |  - |  78.55 | -2.65 |
> | Swin-T  	| LSP | 70 | 71.1  | 16x16  | 79.24  | -1.96 |
> | Swin-T  	| LSP | 79 | 77  | 16x16 | 80.0  | -1.2 |
>
> ---
>
> ## Q5
> It would be great to show some visual results (e.g., attention maps) to prove the effectiveness when compared with other methods.
>
> ## Answer to Q5
> We generated visualization results for attention maps in transformer blocks and added a qualitative comparison with another ViT pruning method SViTE [2]. Following the same visualization setting in [2], we plot out the averaged attention probabilities of all 12 layers and 12 attention heads on 128 images on ImageNet validation set. As shown in Fig. 5 in the appendix, we can see the advantages of our approaches from the visualization results, i.e., the visualized attention maps generated by our approach preserves more information than the coarser structural pruning scheme seen in SViTE. The related figure and detailed analysis for visualization are added in the appendix of the revised manuscript (see Appendix B, Fig. 5). Thanks for this suggestion.
>
> Once again, thank you for your recognition of our work and your valuable advice to allow ous to improve the work.
>
> Best,
>
> Authors
>
> ---
>
>
> ### Reference
> [1] Yu, Lu, and Wei Xiang. "X-Pruner: eXplainable Pruning for Vision Transformers." Proceedings of the IEEE/CVF Conference on Computer Vision and Pattern Recognition. 2023.
>
> [2] Chen, Tianlong, et al. "Chasing sparsity in vision transformers: An end-to-end exploration." Advances in Neural Information Processing Systems 34 (2021): 19974-19988.

---

### Official Review · Reviewer_cWB3 · 2023-11-06

**Soundness:** 3 good
**Presentation:** 3 good
**Contribution:** 2 fair
**Rating:** 5
**Confidence:** 5

**Summary:**

This paper uses a block-structured pruning method to address the resource-intensive issue for ViTs by leveraging more efficient matrix multiplications. By achieving this goal. this paper proposes a novel hardware-aware learning objective that simultaneously maximizes speedup and minimizes power consumption during inference for the block sparsity structure. Besides that, a lightweight algorithm to achieve
post-training pruning for ViTs is provided. DeiT-B and DeiT-S-based experiments are provided.

**Strengths:**

1. Very detailed mathematical conduction.
2. Achieve multiple different block settings in the experimental analysis.
3. Accuracy result looks good compared with the baseline.

**Weaknesses:**

1. The paper claims to achieve hardware-aware learning objective that simultaneously maximizes speedup and minimizes power consumption. But there is no related result evaluation on the hardware side. And there is no hardware setup.
2. The novelty is not clear. Block pruning has been well studied during these years under different block formations and size configurations. Is the novelty mainly for the combination of block pruning with ViTs? If so, what is the main challenge and novelty compared with the previous block pruning work? If the hardware-aware strategy is the main contribution, where are the corresponding hardware configuration results?
3. Only two Deit models are incorporated into the experiments. Not sure about the method's flexibility.

**Questions:**

Please refer to the weakness.

---

> ### Author Response · Authors · 2023-11-18
> **(Q1) Thank you for your recognition and questions! Here is the answer to Question 1. (Part 1/2)**
>
> ## Q1
> The paper claims to achieve hardware-aware learning objective that simultaneously maximizes speedup and minimizes power consumption. But there is no related result evaluation on the hardware side. And there is no hardware setup.
>
> ## Answer to Q1 (Part 1/2)
> We added evaluation on three types of hardware platforms: a RISC-V platform ViTA [1], a DNN-targeted accelerator TPU v3 [2], a GPU platform NVIDIA A100. Results show that our approach is able to bring noticeable improvements for ViT models on the hardware platforms, which demonstrate the effectiveness of our approach.
>
> ---
>
> The first group of evaluation is on a RISC-V platform ViTA [1]. Our approach obtains **5.19x, 4.14x and 1.62x** speedups for DeiT-B, DeiT-S and DeiT-T on ViTA, respectively, without hurting the accuracy. Table 1 below shows the details.
>
> > ### Table 1. Hardware Speedup on VITA
> | Model | FLOPs Remained (%) | Blocksize | Inference Time (ms) | Speedup | Top-1 Accuracy |
> |:------:|:-----:|:-----:|:-----:|:-----:|:-----:|
> | DeiT-B 	| 100 | -   | 16.49  | - | 81.8 |
> | **DeiT-B** 	| **72.8** | **32x32**  | **12.03** | **1.37x** | **80.81** |
> | **DeiT-B** 	| **57** | **32x32**  | **9.59** | **1.72x** | **80.4** |
> | DeiT-B 	| 25.8 | 32x32  | 4.19 | 3.93x | 77.75 |
> | DeiT-B 	| 38.5 | 64x32   |3.15  | 2.61x |  78.1 |
> | DeiT-B 	| 19.8 | 32x64   | 1.59 | 5.19x | 61.42 |
> | DeiT-B 	| 65.9 | 64x64   | 2.71 |  1.52x | 67.67 |
> | DeiT-S 	| 100 | -  | 8.3 |  - | 79.8 |
> | DeiT-S 	| 49.9 | 32x32  | 4.17 | 1.99x | 73.32 |
> | DeiT-S 	| 24.5 | 32x32  | 2.01 |  4.14x | 65.12 |
> | DeiT-S 	| 91.4 | 32x16  | 7.68 |  1.08x | 79.09  |
> | DeiT-S 	| 71.3 | 16x32  | 5.94 |  1.4x | 78.2  |
> | DeiT-T 	| 100 | -  | 4.24 |  - | 72.2 |
> | DeiT-T 	| 61.2 | 16x16  | 2.61 | 1.62x | 69.06 |
>
> With MACs (multiply accumulators) available in various blocksizes, the speedup on ViTA can be achieved as closest to the theoretical target.
>
> ---
>
> In the second group, we also adapt our block-sparse ViTs on Google TPU V3 [2], adopting SCALE-sim [3] to simulate the time cycle. Since TPU v3 only offers 1 type of MAC with block size fixed at 128x128, we expect less speedup than on RISC-V because all smaller-sized blocks from our tested configs (as large as 64x64) within 128x128 must be pruned to skip the computation, which means the hardware effective block-level sparsity is smaller than on VITA. Nevertheless, our approach still obtains noticeable speedups for the ViT models on TPU, bringing about at most **2.57x, 1.54x, 1.02x** speedups for DeiT-B, DeiT-S and DeiT-T, respectively. Table 2 below shows the results.
>
> > ### Table 2. Hardware Speedup on TPU V3
> | Model  | FLOPs Remained (%) | Blocksize | Inference Time (s) | Speedup | Top-1 Accuracy  |
> |:------:|:------------------:|:---------:|:------------------:|:-------:|:---------------:|
> | DeiT-B | 100                | -         | 18.08              | -       | 81.8            |
> |  **DeiT-B** 	| **57** | **32x32**  | **16.74** | **1.08x** | **80.81** |
> | **DeiT-B** 	| **57** | **32x32**  | **16.28** | **1.11x** | **80.4** |
> | DeiT-B | 25.8               | 32x32     | 11.46              | 1.57x   | 77.75           |
> | DeiT-B | 38.5               | 64x32     | 12.62              | 1.43x   | 78.1            |
> | DeiT-B | 19.8               | 32x64     | 7.26               | 2.57x   | 61.42           |
> | DeiT-B | 65.9               | 64x64     | 7.59               | 2.46x   | 67.67           |
> | DeiT-S | 100                | -         | 2.7                | -       | 79.8            |
> | DeiT-S | 49.9               | 32x32     | 2.13               | 1.27x   | 73.32           |
> | DeiT-S | 24.5               | 32x32     | 1.75               | 1.54x   | 65.12           |
> | DeiT-S | 91.4               | 32x16     | 2.7                | 1x      | 79.09           |
> | DeiT-S | 71.3               | 16x32     | 2.62               | 1.03x   | 78.2            |
> | DeiT-T | 100                | -         | 0.00063            | -       | 72.2            |
> | DeiT-T | 61.2               | 16x16     | 0.00061            | 1.02x   | 69.06           |
>
> ### References
> [1] C.Y. Chen et al., “ViTA: A Highly Efficient Dataflow and Architecture for Vision Transformers,” 2024 DATE.
>
> [2] Jouppi, Norman P., et al. "A domain-specific supercomputer for training deep neural networks." Communications of the ACM 63.7 (2020): 67-78.

---

> ### Author Response · Authors · 2023-11-18
> **(Q1) Thank you for your recognition and questions! Here is the answer to Question 1. (Part 2/2)**
>
> In the third group, we deploy on NVIDIA A100 40GB GPU with CUDA 11.8 and evaluate the end-to-end inference time and the runtime power consumption as shown in Table 3., where we also observe a power reduction up to 60.4% on DeiT-B. Power consumption is measured by averaging nvidia-smi’s power meter over an adequate time period.
>
>
>
> > ### Table 3. Hardware Speedup and power consumption on A100 GPU
> | Model  | FLOPs Remained (%) | Blocksize | Batchsize | Inference Time (ms) | Speedup | Power (W) | Top-1 Accuracy  |
> |:------:|:------------------:|:---------:|:---------:|:-------------------:|:-------:|:---------:|:---------------:|
> | DeiT-B | 100                | -         | 4         | 10.76               | -       | 167       | 81.8            |
> | **DeiT-B** 	| **57** | **32x32**  | **4** | **7.73**  | **1.39x** | **152 (91%)** | **80.4** |
> | DeiT-B | 25.8               | 32x32     | 4         | 6.16                | 1.75    | 120 (71.8%)       | 77.75           |
> | DeiT-B | 38.5               | 64x32     | 4         | 6                   | 1.79    | 147 (88%)       | 78.1            |
> | DeiT-B | 19.8               | 32x64     | 4         | 6.11                | 1.76    | 101 (60.4%)      | 61.42           |
> | DeiT-B | 65.9               | 64x64     | 4         | 6.5                 | 1.66    | 122 (73%)      | 67.67           |
> | DeiT-S | 100                | -         | 16        | 11.57               | -       | 195       | 79.8            |
> | DeiT-S | 49.9               | 32x32     | 16        | 8.12                | 1.42    | 160 (82%)      | 73.32           |
> | DeiT-S | 24.5               | 32x32     | 16        | 6.02                | 1.92    | 142 (72.8%)      | 65.12           |
> | DeiT-S | 91.4               | 32x16     | 16        | 10.8                | 1.07    | 192 (98%)      | 79.09           |
> | DeiT-S | 71.3               | 16x32     | 16        | 9.62                | 1.2     | 181 (92.8%)      | 78.2            |
> | DeiT-T | 100                | -         | 256       | 5.12                | -       | 216       | 72.2            |
> | DeiT-T | 61.2               | 16x16     | 256       | 4.15                | 1.23    | 207 (95.8%)      | 69.06           |

---

> ### Author Response · Authors · 2023-11-18
> **(Q2 Q3) Thank you for your recognition and questions! Here is the answer to Question 2,3. (cont'd)**
>
> ## Q2
> The novelty is not clear. Block pruning has been well studied during these years under different block formations and size configurations. Is the novelty mainly for the combination of block pruning with ViTs? If so, what is the main challenge and novelty compared with the previous block pruning work? If the hardware-aware strategy is the main contribution, where are the corresponding hardware configuration results?
>
> ## Answer to Q2
> We agree that block pruning has been studied in previous works especially for language models like BERTs, which also provide a lot of guidance to us. Our approach is not only combining block pruning with ViTs. Actually, the main challenge is how to distribute sparsity under the block-structured pruning scheme for ViTs in a way that both high accuracy and high hardware performance can be achieved, which is absent in prior block-sparse pruning works. Hence, we proposed a novel learning objective that includes hardware-aware constraints to boost the speed and low power consumption for the inference stage. To our best knowledge, our work is the first to present a hardware-aware block-structured pruning for ViTs.
>
> Another challenge of block-structured pruning is the huge search space of optimization, making it a difficult problem to solve. To address this issue, we propose a practical solution to implement our pruning algorithm extremely efficiently despite adopting second-order Taylor approximation.
>
> Finally, to concretely evaluate the hardware efficiency of our method, we added the evaluation results on real hardware platforms to further demonstrate the effectiveness of our approach. Please check our answers to question 1 for the details. We will also modify the manuscript accordingly to clarify more about the novelty of our paper. Thanks a lot for this good point.
>
> ---
>
> ## Q3
> Only two Deit models are incorporated into the experiments. Not sure about the method's flexibility.
>
> ## Answer to Q3
> We added experiments on another **three ViT models: Deit-T, Swin-B and Swin-T**. The results are also promising on those ViT variants. On Deit-T, we obtain around 50% pruning ratio regarding the FLOPs with a 3.14% accuracy drop. On Swin-B, we obtain a 50% FLOPs with the accuracy loss at 3.9% and 1.9% loss at around 71% FLOPs. Table 4 below shows the detailed results.
>
> > ### Table 4. Results on other ViT models
> | Model  | Dataset | Sparsity (%) | FLOPs Remained (%) | Top-1 | Dense | Top-1 diff |
> |:------:|:-----:|:------------:|:-----:|:-----:|:-----:|:-----:|
> | DeiT-T 	| ImageNet | 61.2   | 50   | 69.1 | 72.2  | -3.1 |
> | Swin-B  	| ImageNet | 57.6 | 50 | 79.6  |  83.5 | -3.9 |
> | Swin-T  	| ImageNet | 70 | 71.1 | 79.3 |  81.2 | -1.9 |
>
> Given limited time, we still haven't explored finetuning this two Swin transformers with knowledge distillation as adopted in our existing experiments on DeiTs. With the distillation applied, we believe the performance on Swin transformers could be further boosted in the future.
>
> ---
>
> Looking forward to your kind feedback and please let us know if more experimental discussion is needed.
>
> Best,
>
> Authors.
>
> ---
>
> ### References
> [1] C.Y. Chen et al., “ViTA: A Highly Efficient Dataflow and Architecture for Vision Transformers,” 2024 DATE.
>
> [2] Jouppi, Norman P., et al. "A domain-specific supercomputer for training deep neural networks." Communications of the ACM 63.7 (2020): 67-78.
>
> [3] A. Samajdar, J. M. Joseph, Y. Zhu, P. Whatmough, M. Marina, and T. Krishna, “A Systematic Methodology for Characterizing Scalability of DNN Accelerators using SCALE-Sim,” in 2020 IEEE International Symposium on Performance Analysis of Systems and Software (ISPASS), 2020, pp. 58–68.

---

> ### Comment · Reviewer_cWB3 · 2023-11-22
> **Reply to author**
>
> Thanks for your detailed comparison including the hardware performance. The main concern of mine is the accuracy degradation in the experiments. Most of the pruning results have a huge accuracy degradation compared with the baseline. Even for the results with a 1.02 to 1.03\% speedup, they still have a 1.6 to 3.14\% accuracy drop, which needs tailored optimization. I recommend finding better pruning optimization to overcome this challenge.

---

> > ### Author Response · Authors · 2023-11-22
> > **Thanks a lot for your reply and kind suggestions!**
> >
> > Thanks a lot for paying attention to our response and your kind reply. We have been working towards a more tailored optimization on the ViT models just as you suggested, such as trying different combinations of blocksize and FLOPs targets as well as finetuning strategies. And we already have some promising results on DeiT-Base with improved trade-off between accuracy and hardware speedups, which achieves <1% and 1.4% accuracy loss with 1.37x and 1.72x speedups respectively. We are still working on DeiT-Small and DeiT-Tiny with more pruning optimization and we promise to update the results and revise the manuscript once they are done.
> >
> > We have updated the Table.1,2,3 in our previous response with the newly added DeiT-Base results marked in **Bold**.
> >
> > Best regards,
> >
> > Authors.

---

### Author Response · Authors · 2023-11-22
**Summary of Revision of Manuscript (Many thanks to all reviewers and AC)**

Dear reviewers and AC,

We wholeheartedly appreciate your insightful comments and constructive suggestions provided to help us improve our manuscript.

To best appreciate your comments, we conducted several supplementary experiments and revised the manuscript.
Here is a short list of updates, which are marked in blue in the manuscript.

- Added pivotal works in network pruning in introduction. (Reviewer **yotU**)
- Fixed a typo from "I.d.d." to "I.i.d." in Assumption 1. (Reviewer **yotU**)
- Removed redundant baseline method introduction in Sec. 4.1 "Baseline methods" as already introduced in related works.
- Added Hardware benchmark results in the Appendix. (Reviewer **cWB3, d35y**)
- Added Transfer learning experiments on image segmentation in the Appendix. (Reviewer **d35y**)
- Added Swin Transformer experiments in the Appendix. (**All reviewers**)
- Added visualizations of layerwise pruning ratio distributions and pruning masks in the Appendix. (Reviewer **rMt3**)
- Added attention mask visualizations in the Appendix. (Reviewer **d35y**)
- Substantially revised Related work section (Reviewer **yotU**)

For each individual reviewer comments, we are on the course of giving detailed responses and clarifications to the concerns raised by the reviewers, with pointers to corresponding revisions of the manuscript.

Best regards,

Authors

---

### Author Response · Authors · 2023-11-23
**Summary of Updates (Many thanks to all reviewers and AC)**

Dear Reviewers and AC panel,

We are grateful for the valuable problem raised and the advice given by the reviewers that have helped revise our manuscript. It is glad to see that the merits of our work have been recognized by reviewers **rMt3** and **yotU**. We are summarizing the major points about the paper (post-rebuttal) for a quick understanding of all:

**[Concerns of novelty]** We clarified to our reviewers that the key contributions are (1) the proposed hardware-aware pruning objective tailored to block-sparsity scheme for ViTs and (2) a novel algorithm that significantly reduces the hessian-based pruning criteria. The proposed optimization on semi-structured (block-wise) pruning is appreciated by Reviewer **yotU**.

**[Real Hardware performance]** We provided updated speedups and power consumption benchmark results on **three** different hardware platforms. Table 1,2,3 for Reviewer **cWB3** and Table 3,4,5 for Reviewer **d35y** show empirical evidence of effective speedups and power reductions using our power-constraint method under block-sparse pruning scheme for various ViTs.

**[More architectures needed]** We provided updated experimental results of **DeiT-T** and **Swin transformers**. In Table 2 for Reviewer **rMt3**, **LSP** outperforms counterpart methods on Swin-T. These results prove the effectiveness of UVC across different architectures. More archetecture results, e.g. **T2T** will be added to the manuscript in the future.

**[Evaluate other tasks]** We provided udpated experimental results on two downstream tasks, (1) Cityscapes image segmentation (2) Oxford building image retrieval. In Table 2,7 for Reviewer **d35y**, LSP achieve great performance on downstream tasks, proving the generalizability of LSP pruned models.

**[Lack of visualizations]** We added visualizations and detailed discussions of LSP to the revised manuscript, including (1) layer-wise sparsity allocation results (2) block-sparse patterns obtained from LSP (3) attention probabilities on MHA.

**[Concerns about post-training]** We explained that our post-training recipe is aligned with existing ViTs pruning baselines such as UVC, VTP, SViTE, etc., and we give evidences that post-training is (1) essential to restore model accuracy and (2) cheaper than regular training.

**[Concerns about distillation]** We clarified that the currently comparison with baselines and dense model is adequate as all the model evaluations are performed on output classification token throughout baselines and model. We also provided an ablation study on impact of distillation on the performance in Table 6 for Reviewer **d35y**, showing the distillation is not the main accuracy booster compared to the proposed method.

**[Comapared to uniform pruning]** We gave experimental evidence in Table 1 for Reviewer **rMt3** that shows the proposed accuracy-aware and hardware-aware layer-wise sparsity allocation is superior to vanilla uniform pruning.

**[More organized introduction and related work]** We added the lacking pivotal works in network pruning as well as other important references. We also made substantially reorganized the related work section to provide taxonomy that better fit into the discussed topic in this paper.

**[A few writing concerns]** We made changes in the manuscript such as fixed typo. A new version of pdf is uploaded with the specified modifications.

Best wishes,

Authors

---

### Meta-Review · Area_Chair_sNUc · 2023-12-12

**Metareview:**

The paper introduces a hardware-aware block-structured method for pruning Vision Transformers (ViTs), aiming to improve computational efficiency. Extensive experiments on various hardware platforms demonstrate some speedups and power efficiency. However, the novelty of the approach is questioned due to the extensive existing research in block pruning. While the authors addressed initial concerns about hardware evaluation and model diversity, the paper still faces issues with significant accuracy degradation and unclear discussions on the impact of knowledge distillation. Overall, the paper shows progress in enhancing ViT efficiency but lacks compelling justification for its specific approach over existing methods. It demonstrates improved functionality but falls short in establishing its broader significance or providing clear future research directions, leaving its real-world applicability and impact uncertain.

**Justification For Why Not Higher Score:**

Some limitations pointed out by the reviewers

**Justification For Why Not Lower Score:**

N/A

---

### Decision · Program_Chairs · 2024-01-16

Reject